# Imaging and energetics of single SSB-ssDNA molecules reveal intramolecular condensation and insight into RecOR function

Jason C Bell[1,2†‡], Bian Liu[2,3†§], Stephen C Kowalczykowski[2]*

[1]Graduate Group in Biochemistry and Molecular Biology, University of California, Davis, Davis, United States; [2]Department of Microbiology and Molecular Genetics, University of California, Davis, Davis, United States; [3]Graduate Group in Biophysics, University of California, Davis, Davis, United States

**Abstract** *Escherichia coli* single-stranded DNA (ssDNA) binding protein (SSB) is the defining bacterial member of ssDNA binding proteins essential for DNA maintenance. SSB binds ssDNA with a variable footprint of ~30–70 nucleotides, reflecting partial or full wrapping of ssDNA around a tetramer of SSB. We directly imaged single molecules of SSB-coated ssDNA using total internal reflection fluorescence (TIRF) microscopy and observed intramolecular condensation of nucleoprotein complexes exceeding expectations based on simple wrapping transitions. We further examined this unexpected property by single-molecule force spectroscopy using magnetic tweezers. In conditions favoring complete wrapping, SSB engages in long-range reversible intramolecular interactions resulting in condensation of the SSB-ssDNA complex. RecO and RecOR, which interact with SSB, further condensed the complex. Our data support the idea that RecOR–and possibly other SSB-interacting proteins—function(s) in part to alter long-range, macroscopic interactions between or throughout nucleoprotein complexes by microscopically altering wrapping and bridging distant sites.

**\*For correspondence:** sckowalczykowski@ucdavis.edu

[†]These authors contributed equally to this work

**Present address:** [‡]Department of Biochemistry, Stanford University, Stanford, California, United States; [§]Department of Neuroscience, Johns Hopkins University School of Medicine, Baltimore, United States

## Introduction

Single-stranded DNA (ssDNA) binding protein (SSB) binds rapidly and avidly to ssDNA generated during the normal processes of DNA replication, recombination, and repair (*Meyer and Laine, 1990*). In doing so, SSB protects ssDNA from chemical damage and exonucleolytic degradation, removes secondary structure, and enhances the enzymatic activity of many proteins involved in DNA metabolism (*Shereda et al., 2008*). The extent to which ssDNA is wrapped around a tetramer of SSB is often referred to as a binding mode, defined by the apparent site size or footprint (i.e. nucleotides bound per tetramer). These binding modes are sensitive to salt, temperature, pH, and binding density (*Lohman and Ferrari, 1994*). The cooperativity, i.e. nearest neighbor interactions, of SSB is also altered when SSB binds ssDNA in different binding modes (*Lohman et al., 1986*; *Bujalowski and Lohman, 1987*; *Ferrari et al., 1994*). At low salt concentrations, where ssDNA is partially wrapped around SSB, cooperativity is very high or 'unlimited'. As such, proteins crowd very close to each other along the ssDNA. At higher more physiological salt concentrations, SSB binds in the fully wrapped binding mode and exhibits 'limited' cooperativity, where SSB forms dimers of tetramers (i.e. octamers) along the ssDNA (*Bujalowski and Lohman, 1987*; *Lohman and Ferrari, 1994*).

Early electron microscopic visualization of SSB-coated ssDNA revealed a beads-on-string structure similar to those observed for nucleosomes bound to dsDNA (*Chrysogelos and Griffith, 1982*). These structures are observed at a low binding density of SSB; however, at higher binding densities, the structures form smooth, contoured nucleoprotein complexes that are condensed relative to the

**eLife digest** Chromosomes consist of two strands of DNA that are intertwined as a helix. These strands can peal apart to form single-stranded DNA before the DNA is copied and for other processes in cells. Single-stranded DNA can also form if double-stranded DNA is damaged by harmful radiation or chemicals so that only one strand can be copied or when the damaged strand is selectively degraded by enzymes during the course of repair.

Proteins called single-stranded binding proteins (or SSBs for short) bind to single-stranded DNA molecules to protect them. A molecule of single-stranded DNA wraps around a group of four SSB proteins (known as a tetramer). The degree to which DNA is wrapped around the SSB tetramer depends on the environmental conditions. For example, in the presence of high levels of salt—which is typical inside cells – single-stranded DNA wraps around all four subunits of the SSB. However, at lower salt levels, the DNA only wraps around some of the units in the SSB tetramer.

A process called recombination can repair breaks in DNA. During this process, a broken DNA molecule that contains single-stranded DNA can pair with a matching (or complementary) strand from an intact double-stranded DNA molecule that carries an identical genetic sequence. A protein called RecO helps to anneal two complementary DNA strands together with the help of the RecR protein. However, for RecR and RecO to achieve this task, they need to work together with the resident SSB proteins that occupy single-stranded DNA. How they find matching sequences when SSB proteins are in the way is not clear.

Bell et al. used techniques called TIRF microscopy and single-molecule force spectroscopy to directly observe how SSB from the bacterium *E. coli* binds to and coats individual molecules of single-stranded DNA. The experiments show that when the levels of salt increase, single-stranded DNA that is coated with SSB proteins becomes compacted and the length of the DNA molecules decreases, a process referred to as 'intramolecular condensation'. Bell et al. found that condensation occurred because two SSB tetramers that are associated with different regions of the single-stranded DNA interact to form stable 'octamers'.

In the presence of RecO and RecR, the single-stranded DNA compacted even further. Bell et al. propose that these recombination proteins act as a scaffold to bring together distant partner sites of single-stranded DNA. This condensation allows two DNA sequences that can be far apart in the cell to find one another more quickly. The next challenge is to understand how the matching regions of single-stranded DNA are identified, and what causes the SSBs to move to allow other repair proteins to gain access to the DNA.

contour length of the corresponding dsDNA (*Griffith et al., 1984*; *Hamon et al., 2007*). High-resolution atomic force microscopy (AFM) imaging of spread SSB-coated ssDNA formed in low and high salt, measured approximately a twofold difference between the contour lengths of the nucleoprotein complexes. This difference in contour length was proposed to reflect the partially wrapped $SSB_{35}$ and fully wrapped $SSB_{65}$ binding modes, corresponding to a site size of 35 and 65 nucleotides, respectively (*Hamon et al., 2007*). It is worth noting that an additional, intermediate binding mode, $SSB_{55}$, was also observed in direct binding experiments (*Lohman and Overman, 1985*; *Bujalowski and Lohman, 1986*).

SSB has been studied extensively using single-molecule FRET on short oligonucleotide substrates (*Roy et al., 2007*, *2009*; *Zhou et al., 2011*); however, relatively little is known about the more complex dynamics of the SSB-coated ssDNA nucleoprotein fiber that forms on the extensive regions of ssDNA during DNA unwinding, resection, and replication. These ssDNA regions can range from a few hundred to tens of thousands of nucleotides in length. More than a dozen proteins interact directly with SSB via its short, unstructured C-terminal tail (*Shereda et al., 2008*; *Wessel et al., 2013*; *Bhattacharyya et al., 2014*). In the absence of interaction partners or ssDNA, this unstructured peptide tail interacts with the subunits within the SSB tetramer (*Kozlov et al., 2010a*). This inter-subunit allostery contributes to the complex, cooperative nature of SSB binding to ssDNA. It has been proposed that the binding modes of SSB might be modulated in vivo for differential roles during ssDNA processing. Direct evidence for such modulation remained elusive for many years (*Shereda et al., 2008*); however, recent work has shown that PriC remodels the SSB-ssDNA complex

to create a DNA structure competent for DnaB loading during replication restart (*Wessel et al., 2013*) and that PriA modulates the SSB-ssDNA complex to expose a potential replication initiation site (*Bhattacharyya et al., 2014*).

RecO catalyzes the annealing of complementary strands of ssDNA even in the presence of SSB, which otherwise kinetically blocks annealing (*Kantake et al., 2002*); in this regard, perhaps RecO is mimicking the action of PriA (*Bhattacharyya et al., 2014*). This annealing activity is essential for RecA-independent, homology-directed DNA repair that proceeds through the single-strand annealing (SSA) pathway (*Kantake et al., 2002*). RecO also stimulates RecA-dependent homologous recombination by acting with RecR and RecF to promote RecA filament assembly (*Umezu et al., 1993*; *Morimatsu and Kowalczykowski, 2003*; *Handa et al., 2009*). RecR, which does not bind to ssDNA, dsDNA, or SSB, binds to RecO and enhances the affinity of RecO for ssDNA-bound SSB (*Umezu et al., 1993*; *Umezu and Kolodner, 1994*); however, neither RecO nor RecOR are capable of physically displacing SSB from ssDNA (*Umezu and Kolodner, 1994*; *Ryzhikov et al., 2011*). When RecR is bound to RecO, it partially inhibits the annealing activity of RecO but stimulates both the rate of RecA nucleation and filament growth on SSB-coated ssDNA (*Kantake et al., 2002*; *Bell et al., 2012*; *Morimatsu et al., 2012*). As RecA does not interact with SSB, RecO, or RecR (*Umezu et al., 1993*), this activity must proceed through a RecOR-induced conformational change in the SSB-ssDNA complex (*Ryzhikov et al., 2011*; *Zhou et al., 2011*).

Using both direct visualization of SSB-coated ssDNA and single-molecule force spectroscopy, we observed the reversible intramolecular condensation of single SSB-coated ssDNA fibers. The extent of this intramolecular condensation increases with salt concentration, but exceeds the expected extent of condensation based on most previous measurements of SSB-ssDNA complexes (*Chrysogelos and Griffith, 1982*; *Hamon et al., 2007*). We also observe RecO-induced condensation of the SSB-ssDNA complex, as well as long-range intramolecular bridging in the presence of both RecO and RecR. We propose that the nature of this condensation is due to the ability of SSB to interact with distant sites along the ssDNA—either through dimerization of SSB tetramers or through the partial wrapping of distant ssDNA sites on a single SSB protomer—and that one role of RecOR is to enhance these distant interactions, which in turn would facilitate annealing of complementary strands. Our observations raise the possibility that the microscopic changes in ssDNA-binding modes observed for SSB cause macroscopic condensation (or de-condensation) of the nucleoprotein fiber that, in turn, might regulate access to ssDNA.

## Results

### Single molecules of SSB-coated ssDNA reversibly condense in response to increasing salt concentration

We previously described a fluorescent biosensor for ssDNA derived from an engineered mutant, SSB$^{G26C}$, wherein a fluorophore was conjugated to the protein using Alexa Fluor 488 maleimide to produce SSB$^{AF488}$ (*Dillingham et al., 2008*). This protein maintains a high, albeit attenuated, affinity for ssDNA (*Bell, 2011*; *Bell et al., 2012*). SSB$^{AF488}$-ssDNA nucleoprotein complexes were formed by first denaturing bacteriophage λ genomic dsDNA that had been biotinylated at the 3′-terminated ends using DNA polymerase (*Figure 1A*). The denatured DNA was mixed with buffer containing SSB$^{AF488}$, attached to a glass surface functionalized with streptavidin, extended using flow within a microfluidic chamber, and visualized using total internal reflection fluorescence (TIRF) microscopy (*Figure 1B*). When the concentration of sodium acetate (NaOAc) was increased during buffer exchange at a constant flow rate and a constant concentration of fluorescent SSB, the length of single molecules of SSB$^{AF488}$-coated ssDNA shortened (*Figure 1C*, *Video 1*, and *Figure 1D*); however, the fluorescent intensity of individual molecules remained constant (*Figure 1E*, *Video 1*, and *Figure 1—figure supplement 1*), indicating that the protein had not dissociated, but rather redistributed, along the ssDNA molecule. In contrast, when the SSB$^{AF488}$ was exchanged for unmodified wild type SSB, which has a higher affinity for ssDNA, the fluorescence rapidly decreased as SSB$^{AF488}$ was displaced from the ssDNA (*Figure 1F*).

High resolution imaging of SSB-coated ssDNA, using electron microscopy (EM) and AFM, had previously observed that the length of the nucleoprotein fiber is dependent on the buffer condition in which the complex is formed (*Chrysogelos and Griffith, 1982*; *Hamon et al., 2007*). However, we were perplexed by the observation that the amount of protein—indicated by the total fluorescent intensity—along the ssDNA remained essentially unchanged during each salt-jump transition, despite the fact that the length had changed substantially.

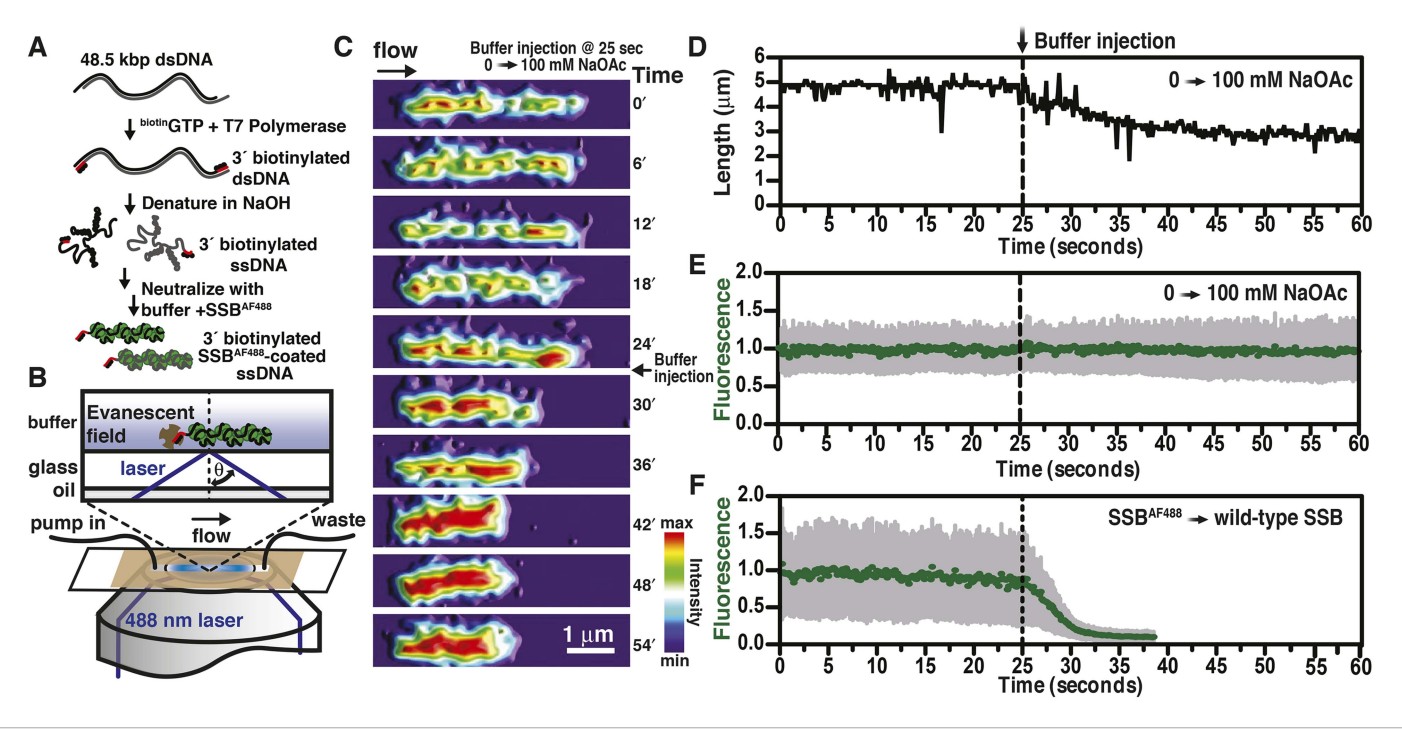

**Figure 1**. Visualization of salt-induced intramolecular condensation of single molecules of *SSB*[AF488]-ssDNA complexes. (**A**) Bacteriophage λ dsDNA (48.5 kbp) was biotinylated, denatured, coated with SSB[AF488], and then (**B**) attached to a streptavidin-coated glass coverslip of a microfluidic chamber where it was extended by buffer flow for direct imaging using total internal reflection fluorescence (TIRF) microscopy. (**C**) A montage of frames from a video recording the change in length of a single molecule of SSB[AF488]-coated single-stranded DNA (ssDNA) upon increasing [NaOAc] from 0 to 100 mM. The frames were rendered into a topographical intensity map. Time zero corresponds to the time at which the pump was turned on. The dead time of the experiment was ~25 s due to the liquid volume in the lines between the syringe valve and the microfluidic chamber. (**D**) The length of the molecule in panel **C** during the change in salt from 0 mM to 100 mM NaOAc was measured for each frame and is plotted as a function of time. The dotted line represents the injection of the buffer into the microfluidic flow chamber. (**E**) The fluorescence intensity of the molecule in panel **C** was also measured for each frame and is plotted as a function of time. (**F**) The fluorescence intensity of a single molecule of SSB[AF488]-coated ssDNA is plotted as function of time during a similar experiment where SSB[AF488] was exchanged for wild-type, unlabeled SSB. The decrease in fluorescence intensity corresponds to the displacement by wild-type SSB, which has higher affinity for ssDNA than SSB[AF488]. The fluorescence intensity (green circles) was determined by the mean pixel intensity of region of interest (ROI), and the gray error bars are the standard deviation of the pixels within the ROI.

The following figure supplement is available for figure 1:

**Figure supplement 1**. Intensity and length measurements during salt-induced intramolecular condensation.

Stopped-flow kinetic studies have previously demonstrated that SSB tetramers can transfer between ssDNA molecules without proceeding through a free protein intermediate (*Kozlov and Lohman, 2002a, b*) and single-molecule experiments have directly demonstrated that SSB tetramers diffuse rapidly on ssDNA and can 'hop or jump' across long distances of ssDNA via intersegmental transfer (*Roy et al., 2009*; *Zhou et al., 2011*; *Lee et al., 2014*). To distinguish between the intramolecular redistribution of SSB along the ssDNA in cis vs dissociation balanced with rebinding during the transition, we asked whether the salt-induced condensation of single molecules might be reversible in the absence of free protein. To address this possibility, SSB-coated ssDNA was tethered in a flow cell and extensively washed with buffer to remove free protein (~100–200 volumes of the flow chamber). An injection loop was then used to transiently pulse the tethered molecules with buffer containing either 100 or 400 mM NaOAc, followed by a sufficient volume of buffer to remove the injected salt. When the salt concentration was raised to 100 mM NaOAc, the flow-extended molecules compacted to ~60% of the length in the absence of salt (*Figure 2A, Video 2*), and then returned to the previously extended length when the salt was removed from the flow chamber. Similarly, this condensation was also observed when 400 mM NaOAc was used (*Figure 2B, Video 3*); however, the

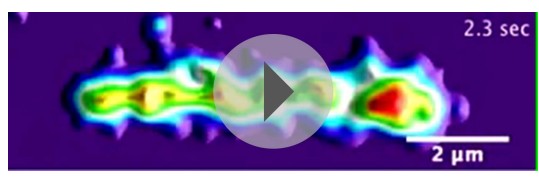

**Video 1.** Salt-induced intramolecular condensation of SSB[AF488]-ssDNA. Video recording of a single molecule of SSB[AF488]-coated ssDNA, imaged using TIRF microscopy, upon increasing [NaOAc] from 0 to 100 mM. The video frames were rendered into a topological intensity map. Time zero corresponds to the time at which the pump was turned on. The dead time of the experiment was approximately 25 s due to the volume in the lines between the syringe valve and the microfluidic chamber. The molecule in the video corresponds to the molecule presented in *Figure 1*, panels **C–E**

extent of the condensation was greater, wherein the molecules shortened to ~12% of the flow-extended length in the absence of salt. In both experiments, the molecules were dimmer at the end of the experiment (*Figure 2—figure supplement 1*), where approximately 20% of the SSB dissociated in the 0→100 transition, and ~60% dissociated in the 0→400 transition.

## The extent of intramolecular condensation of SSB-coated ssDNA exceeds expectations based on simple wrapping or binding-mode transitions

SSB[AF488] is particularly suitable for single molecule measurements due to its relative photo-stability, whereas an alternative biosensor derived from fluorescein-5-maleimide, SSB[f], is particularly suitable as an ensemble ssDNA-biosensor due to the large, linear increase in fluorescence upon ssDNA binding (*Dillingham et al., 2008*; *Bell, 2011*). To determine whether the measured lengths of individual SSB[f]-coated ssDNA complexes were correlated with the DNA-binding modes of SSB, we measured the stoichiometry of SSB[f] binding to poly(dT) using ensemble fluorescence measurements. Poly(dT) was titrated into a fixed concentration of SSB[f] at various concentrations of NaOAc, and the data were fit to a two-segment line to determine the apparent site size, which reflects the extant binding mode at each salt concentration (*Figure 3A*). The observed site size increased from ~43 to ~70 nucleotides per SSB tetramer over the range of salt concentrations tested as expected; however, we noted that the amplitude of the fluorescence enhancement increased dramatically with salt concentration, indicating the molecular environment of the fluorophore was altered (*Figure 3B*). In addition to the stoichiometric titrations performed by adding ssDNA to a fixed concentration of SSB[f], we performed so-called 'salt back-titrations' to determine the concentration at which SSB[f] dissociates from ssDNA. When pre-formed complexes of SSB[f]-poly(dT) were titrated with an increasing concentration of salt (*Figure 3—figure supplement 1*), we observed an initial sharp increase in the fluorescence corresponding to the amplitudes from our direct titrations shown in *Figure 3A*. The fluorescence peaked between 200 and 400 mM NaOAc, and was followed by a shallow, linear decrease until the concentration reaches approximately 2M NaOAc, where the fluorescence intensity exhibited a sharp decrease due to dissociation (*Figure 3—figure supplement 1*). The midpoint of this sharp transition corresponds to the so-called, 'salt-titration midpoint', where ~50% of the complex is dissociated (*Kowalczykowski et al., 1981*; *Newport et al., 1981*). The salt-titration midpoint for this experiment shows that ~2 M NaOAc in the presence of 5 mM Mg(OAc)$_2$ is required to dissociate half of the protein from the DNA.

Individual SSB[f]-ssDNA complexes were also visualized with TIRF microscopy at increasing concentration of NaOAc, and it was evident that the length of the nucleoprotein fibers decreased as the NaOAc concentration increased (*Figure 3C*). Because we initially considered that the change in length might simply correspond to the change in the salt-dependent binding mode, we plotted the apparent site size determined from the titrations shown in *Figure 3A*, as a function of increasing [NaAOc] (*Figure 3D*) and compared this to the average length of SSB[f]-ssDNA complexes (*Figure 3E*), measured from images such as those in *Figure 3C* and more thoroughly represented in *Figure 3—figure supplement 2*. This comparison shows that the site size changes approximately ~1.7-fold (43 nts to 70 nts) over this range whereas the length of the SSB-ssDNA molecules changes approximately ~13-fold (from ~6.5 μm to ~0.5 μm). We note that the apparent site size of SSB can vary depending on the ssDNA used owing to exclusion of SSB from regions capable of forming stable secondary structure (*Lohman and Overman, 1985*); however, the reported change in site size for natural M13 ssDNA is only ~2.2-fold (from 35 to 77) over a comparable range (1 mM to 300 mM NaCl), which is insufficient to account for our observations.

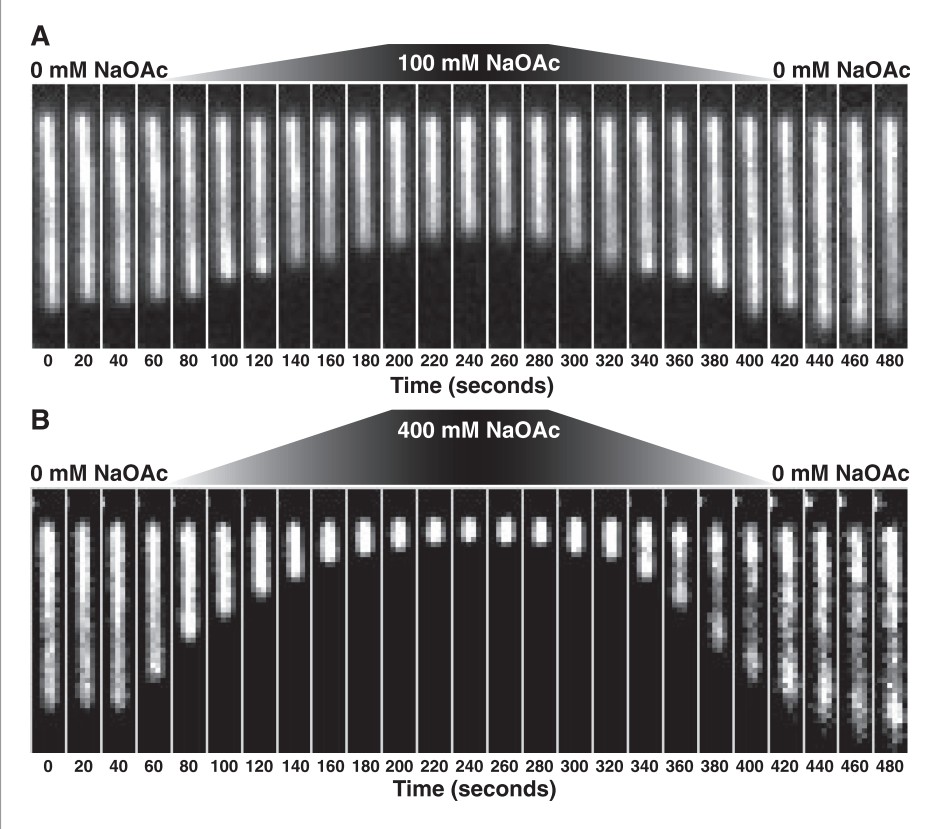

**Figure 2**. The length change upon salt-induced condensation of SSB[AF488]-coated ssDNA is nearly reversible in the absence of free SSB protein. (**A**) A montage of frames from a video recording of a single molecule of SSB[AF488]-coated ssDNA contracting in length as the salt concentration is increased from 0 to 100 mM NaOAc, and then subsequently reduced back to zero, conducted in the absence of free SSB[AF488]. The flow cell was extensively washed with buffer to remove free SSB protein from the flow cell before beginning the experiment. Video recording began when the pump was turned on, requiring ~40-50 s for the dead volume to be flushed from the lines to the flow chamber. (**B**) Same as in (**A**), except the salt concentration was increased from 0 mM NaOAc to 400 mM NaOAc and then back to zero. Each frame of the montages is one micron wide. SSB[AF488] was omitted from both of the high salt washes and from the 0 mM wash. Flow is from top to bottom in each image.

The following figure supplement is available for figure 2:

**Figure supplement 1**. SSB[AF488] partially dissociates from ssDNA during salt transitions in the absence of free protein.

---

If SSB[f] dissociated from ssDNA during the salt transitions, then formation of secondary structure could explain the additional compaction; however, when we measured the intensity of individual SSB[f]-ssDNA fibers at each salt concentration (*Figure 3—figure supplement 3*), we see an increase in fluorescence intensity similar to—and in good agreement with—the titration performed with poly(dT) in *Figure 3A,B*, and *Figure 3—figure supplement 1*. We note that the increase in fluorescence observed in *Figure 3—figure supplement 3* is due to the environmental sensitivity of SSB[f], and should not be confused with the results from *Figure 1*, where we used SSB[AF488]. Owing to the complex changes in fluorescence upon ssDNA binding, we cannot completely rule out the possibility that protein partially dissociates from the SSB[f]-ssDNA; however, we see no evidence of significant net dissociation in our assay in the presence of free protein. In the absence of free protein in solution, dissociation is apparent during the salt transitions (*Figure 2—figure supplement 1*), indicating that the net constant intensity we observe in *Figure 1* and *Figure 1—figure supplement 1* is maintained by mass action and rapid re-binding and redistribution of SSB along the ssDNA polymer.

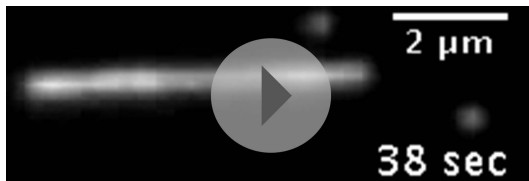

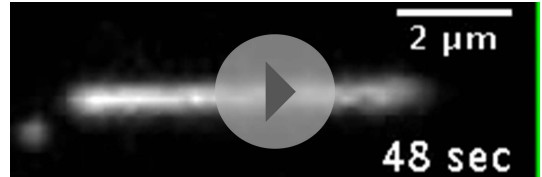

**Video 2.** Condensation of SSB[AF488] in the absence of free protein during a transient increase from 0 to 100 mM NaOAc. Video recording of a single molecule of SSB[AF488]-coated ssDNA contracting in length as the salt concentration is increased from 0 to 100 mM NaOAc, and then subsequently reduced back to zero mM, conducted in the absence of free SSB[AF488]. The flow cell was extensively washed with buffer to remove free SSB protein before beginning the experiment. Video recording began when the pump was turned on, requiring ~40-50 s for the dead volume to be flushed from the lines to the flow chamber. SSB[AF488] was omitted from both of the high-salt washes and from the 0 mM wash. The video corresponds to the molecule presented in *Figure 2A* and *Figure 2—figure supplement 1*, panels **B**, **C**.

**Video 3.** Condensation of SSB[AF488] in the absence of free protein during a transient increase from 0 to 400 mM NaOAc. Video recording of a single molecule of SSB[AF488]-coated ssDNA contracting in length as the salt concentration is increased from 0 to 400 mM NaOAc, and then subsequently reduced back to zero mM, conducted in the absence of free SSB[AF488]. The flow cell was extensively washed with buffer to remove free SSB protein before beginning the experiment. Video recording began when the pump was turned on, requiring ~40-50 s for the dead volume to be flushed from the lines to the flow chamber. SSB[AF488] was omitted from both of the high-salt washes and from the 0 mM wash. The video corresponds to the molecule presented in *Figure 2B* and *Figure 2—figure supplement 1*, panels **D**, **E**.

To further assess the condensation state of SSB-coated ssDNA at approximately physiological ionic conditions, we measured the SSB[f]-ssDNA lengths in the presence of the divalent cation magnesium, which is known to affect SSB binding modes (*Bujalowski et al., 1988*). $Mg(OAc)_2$ induces a large condensation of nucleoprotein fibers that plateaus between 1 and 2 mM, and results in complexes that are as short as those produced with the much higher monovalent salt concentrations (*Figure 3F* and *Figure 3—figure supplement 4*). *Escherichia coli* maintains its intracellular osmolality by adjusting the intracellular concentration of glutamate, which ranges from ~30 to 260 mM when cells are grown in media containing from ~100 to 1100 mM solute (*Richey et al., 1987*). Therefore, we also titrated sodium glutamate (NaGlu) in the presence and absence of 1 mM $Mg(OAc)_2$, which is within the range of the measured intracellular free magnesium ion concentration (1–2 mM) (*Alatossava et al., 1985*). At low concentrations of NaGlu (*i.e.* below 100 mM), the condensation was dominated by the presence of 1 mM $Mg(OAc)_2$; however, the observed condensation became dominated by NaGlu at higher concentrations (*Figure 3G* and *Figure 3—figure supplement 5 and 6*). In the absence of $Mg(OAc)_2$, there was a log-linear decrease in length with increasing concentrations of NaGlu, similar to our observation for NaOAc. Extrapolating from this data, the shorter, more condensed molecules that we observe likely represent the physiologically relevant condensation state of the SSB-ssDNA complex as estimated by the in vivo concentration and composition of salts.

## Force spectroscopy of single molecules of ssDNA and SSB-coated ssDNA reveals a nearly complete relief of hysteresis in SSB-ssDNA unfolding transitions

To further explore the intramolecular condensation of SSB-coated ssDNA, we used a magnetic tweezer instrument to generate force-extension curves (*Gosse and Croquette, 2002*; *Meglio et al., 2009*). In particular, we reasoned that force spectroscopy would enable us to distinguish between intramolecular collapse owing to secondary structure formation and exclusion of SSB versus intrinsic, protein-mediated folding of the SSB-ssDNA molecule. We further reasoned that because SSB[f] is a modified variant of SSB (*Dillingham et al., 2008*; *Bell, 2011*), we could not exclude the possibility that a component of the intramolecular condensation might be due to the fluorescent adduct. This concern prompted us to assess the condensation state of single molecules of ssDNA coated with wild-type, unmodified SSB. Briefly, a ~13.5-kbp DNA substrate with unique flanking restriction sites was PCR amplified from bacteriophage λ DNA. Molecular 'handles' were made by PCR amplification of

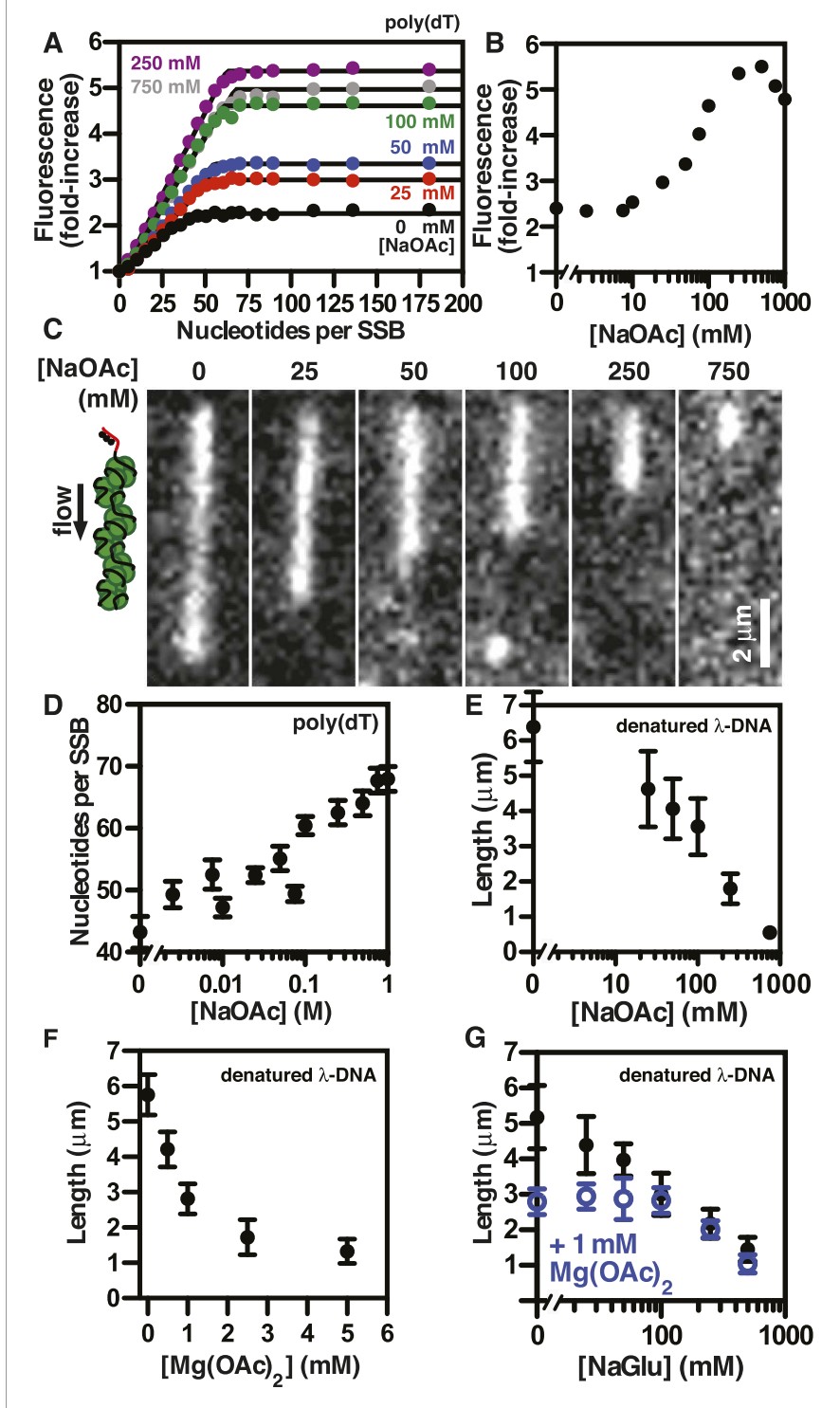

**Figure 3**. The extent of SSB-ssDNA condensation is greater than anticipated based on known ssDNA-wrapping transitions. (**A**) Poly(dT) was titrated into 100 nM SSB$^f$ (tetramer) and the average fluorescence enhancement of SSB$^f$ from three titrations was plotted as a function of ratio of poly(dT) to SSB tetramer. The data were fit to a two-segment line, where the breakpoint is the stoichiometric endpoint of the titration corresponding to the site size of SSB$^f$. (**B**) The amplitude of the fluorescence enhancement from the titrations performed in *Figure 3A* was plotted as a function of [NaOAc]. The error is smaller than the symbols. A larger number of titrations are shown here than in panel **A** to prevent panel **A** from being overcrowded; each fold-increase was determined by a full stoichiometric titration where each titration was completely and fully saturated. (**C**) Representative images of single molecules of SSB$^f$-coated ssDNA at increasing

*Figure 3. continued on next page*

*Figure 3. Continued*

[NaOAc] indicated. (**D**) The apparent binding site size (black circles, ± error of the fits from panel **A** determined from the titrations performed in panel **A** were plotted as a function of salt concentration. (**E**) Length of SSB$^f$-coated ssDNA molecules plotted as a function [NaOAc] (N = 213). (**F**) Length of SSB$^f$-coated ssDNA plotted as a function of [Mg(OAc)$_2$] (N = 156) and (**G**) as a function of [NaGlu] in the absence (black, closed circles, N = 205) and presence (blue, open circles, N = 214) of 1 mM Mg(OAc)$_2$. Unless otherwise indicated, all error bars represent standard deviation and when not visible were smaller than the symbols.

The following figure supplements are available for figure 3:

**Figure supplement 1**. Salt back-titrations to determine the concentration at which SSBf dissociates from ssDNA.

**Figure supplement 2**. Length distributions of single molecules of SSB$^f$-coated ssDNA as a function of [NaOAc].

**Figure supplement 3**. Intensity of SSB$^f$-ssDNA molecules as a function of [NaOAc].

**Figure supplement 4**. Length distributions of single molecules of SSB$^f$–coated ssDNA as a function of [Mg(OAc)$_2$].

**Figure supplement 5**. Length distributions of single molecules of SSB$^f$-coated ssDNA as a function of [NaGlu].

**Figure supplement 6**. Length distributions of single molecules of SSB$^f$-coated ssDNA as a function of [NaGlu] in the presence of 1 mM Mg(OAc)$_2$.

two fragments (∼2 kbp each) using pUC19 as a substrate in the presence of modified nucleotides. One fragment was amplified in the presence of DIG-dUTP, while the other was amplified in the presence of biotin-dGTP. These PCR fragments were ligated to the ends of the 13.5 kbp substrate (*Figure 4A*, leftmost cartoon). The DNA substrate was alkali denatured and then neutralized with buffer (*Figure 4A*, center cartoon). This resulted in ssDNA with one end that could be attached to the glass surface of a microfluidic cell that had been functionalized with anti-DIG antibodies, and the other end could be attached to a streptavidin-coated paramagnetic bead (*Figure 4A*, rightmost cartoon). Paramagnetic beads and ssDNA that were not attached to the surface were flushed from the flow cell. Force-extension measurements of individual molecules were performed first in the absence of SSB at increasing salt concentrations. The salt was flushed from the flow cell, and measurements were then repeated in the presence of SSB, again at increasing concentrations of salt.

The extension of the ssDNA was measured by incrementally increasing and then decreasing the distance between the magnet and the surface, thereby decreasing and increasing the force, respectively. When SSB was bound to the ssDNA, both increasing and decreasing the force kinetically perturbed the length of the SSB-ssDNA complex with apparent exponential kinetics (*Figure 4B*, inset). Care was taken to insure that the length measurements were made only after the ssDNA or SSB-ssDNA complex was in equilibrium at each force (*Figure 4B*). Force-extension curves of individual molecules of ssDNA were obtained by first decreasing force (dashed lines) and then increasing force (solid lines) at increasing concentrations of NaOAc in the absence (*Figure 4C*) and presence of SSB (*Figure 4D* and *Figure 4—figure supplement 1*). The force-extension curves of ssDNA alone show that the force required to stretch ssDNA also increases with respect to increasing concentrations of NaOAc (*Figure 4C*). The force-extension curves for ssDNA alone at salt concentrations greater than 25 mM NaOAc demonstrate hysteresis (i.e. the force-extension curves obtained by increasing and decreasing force do not overlap) due to the formation of secondary structure in the ssDNA (*Bosco et al., 2014*) see also, (*Zhang et al., 2001*; *Saleh et al., 2009*). Interestingly, this hysteresis was absent in the force-extension curves measured in the presence of SSB (*Figure 4D*), except at 750 mM NaOAc, the highest salt concentration that we used in our magnetic tweezer experiments. This modest hysteresis at the highest concentration of NaOAc is consistent with our interpretation that the decrease in the amplitude of the fluorescence enhancement measured during equilibrium titrations with SSB$^f$ is due to partial dissociation of SSB at concentrations of NaOAc greater than 750 mM (*Figure 3A,B*). More importantly, the absence of hysteresis in the presence of SSB is strong evidence that the structures are always at equilibrium, and that there are no kinetically trapped intermediates,

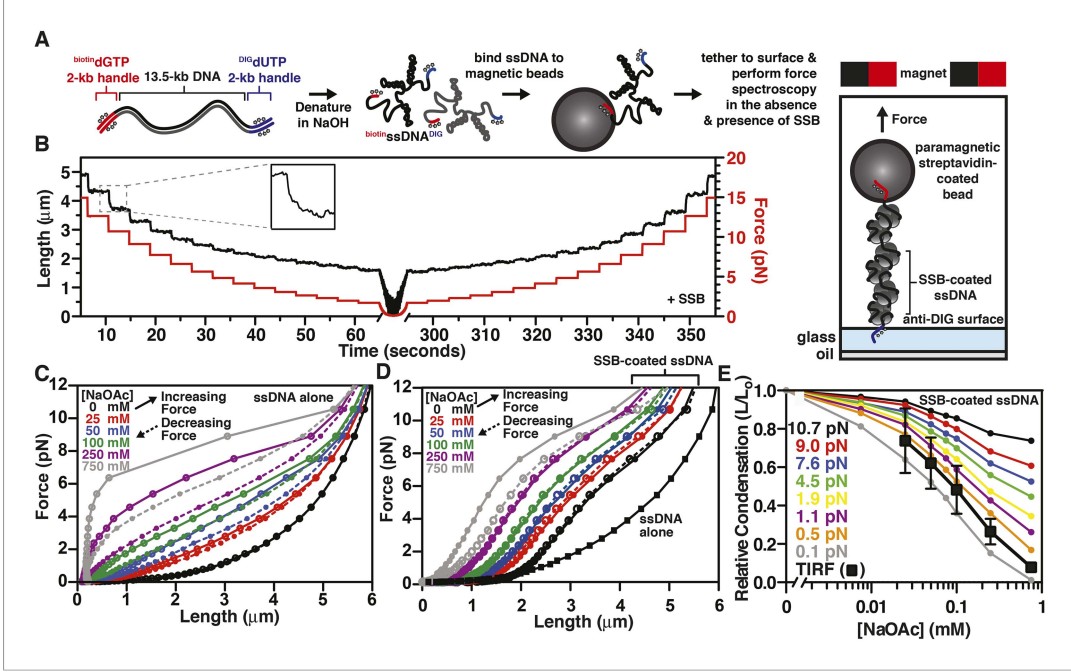

**Figure 4**. The binding of SSB eliminates hysteresis from the force-extension behavior of ssDNA measured by single-molecule magnetic tweezer force spectroscopy (**A**) A DNA substrate was made by ligating biotin- and DIG-containing 'handles' (i.e. ~2-kb products from PCR containing biotin-dGTP or DIG-dUTP) to the flanking ends of a 13.5-kb DNA substrate. The ligated product was then alkali denatured, attached to magnetic streptavidin-coated beads, and then tethered to a glass surface coated with anti-DIG antibodies within a flow chamber. When present, SSB was added to the flow chamber and bound to the ssDNA in situ. (**B**) A typical time trace of a single molecule of SSB-coated ssDNA during a force-extension experiment. The length was determined at each force applied after the molecule had reached equilibrium (inset). (**C**) The force-extension and relaxation relationship of a single ssDNA molecule is shown in the absence of SSB at increasing concentrations of NaOAc. The plot shows length measurements made while both decreasing (dashed lines) and increasing force (solid). (**D**) The force-extension relationship of ssDNA in the presence of 200 nM SSB at increasing concentrations of NaOAc. (**E**) The relative condensation ($L/L_0$, where $L_0$ is the length of SSB-coated ssDNA in the absence of salt) of the molecules measured in panel D were plotted as a function of salt concentration for each applied force and compared to the relative condensation of molecules measured in direct visualization (TIRF) experiments from *Figure 3C,E*.

The following figure supplement is available for figure 4:

**Figure supplement 1**. Force extension curve of SSB-coated ssDNA at low force.

neither short range nor the more likely long range random interactions (*Schaper et al., 1991*), that would contribute to non-equilibrium behavior. We therefore infer that the intramolecular condensation that we observe is intrinsic to a reversible equilibrium folding of the SSB-ssDNA complex.

The length of SSB-coated ssDNA from *Figure 4D* was normalized to that in the absence of salt in order to determine the relative salt-induced condensation at each force measurement. This relative condensation was calculated by dividing the length, L, by $L_0$ (i.e. $L/L_0$), where $L_0$ is the length of the SSB-coated ssDNA in the absence of salt (0 mM NaOAc) for each applied force. The relative condensation was then re-plotted as a function of increasing NaOAc concentration for each force measurement (*Figure 4E*). As the concentration of NaOAc increases, the length of the SSB-coated ssDNA monotonically decreases, or condenses, (*Figure 4E*); however, the extent of the condensation is dependent on the force applied. We also compared the salt-dependent length changes measured using magnetic tweezers (*Figure 4D,E*) to the salt-dependent length changes observed in the direct visualization experiments using TIRF microscopy (*Figure 3C,E*) by calculating the relative condensation, $L/L_0$, to permit a normalized comparison of the different lengths of ssDNA used in each experiment. Since the salt-induced condensation of the molecules measured in the TIRF

experiments (*Figure 4E*, black squares) is most similar to the relative condensation observed in the magnetic tweezer experiments that were performed at forces between 0.1 and 0.5 pN (*Figure 4E*, orange and gray circles), we estimate that this is approximately the shear force applied by flow during TIRF microscopy within our microfluidic cells. Importantly, because the salt-induced condensation of the SSB$^f$-ssDNA complex is in qualitative agreement with the data obtained with wild-type SSB, we conclude that the salt-induced condensation observed in *Figure 3* and *Figure 4D,E* is dominated by intramolecular bridging in cis, mediated by oligomers of SSB, rather than any property of the modified SSB$^f$ (*see Discussion*).

## The SSB-ssDNA complex is a nearly isoenergetic landscape, relative to unstructured ssDNA, at physiological salt concentrations

Force spectroscopy directly measures the work required to extend a molecule, and thus can be used determine the energetic consequences of protein binding to DNA (*Liphardt et al., 2002*). The integrated area under a force-extension curve is the work performed on—or absorbed by—the polymer that is being stretched (*Figure 5*). We therefore integrated the area under the force-extension curves, relative to the area for ssDNA alone in the absence of salt, to measure the change in energy of the interrogated molecule due to the presence of salt (*Figure 5—figure supplement 1A*) and/or SSB (*Figure 5—figure supplement 1B*). The measured change in energy ($\Delta E$), in units of pN•nm, was then converted to units of $k_BT$ (i.e. the Boltzmann constant multiplied by the absolute temperature) using the relationship $k_BT \sim 4.1$ pN•nm at 25℃, which corresponds to the energy contribution from thermal fluctuations (*Nelson, 2004*). The energy of salt-induced stabilization of ssDNA secondary structure is apparent in the deviation between the results obtained for pulling (*Figure 5A*, increasing force, black filled circles) and relaxing (*Figure 5A*, decreasing force, open circles) ssDNA in the absence of SSB. In the presence of SSB, the change in energy with respect to salt concentration approximately paralleled the behavior of ssDNA being relaxed from high force to low (*Figure 5A*, compare red circles, red line with open black circles, dashed line), where long range ssDNA secondary structure does not contribute energetically. Because there was no hysteresis for SSB-ssDNA complexes (*Figure 4D*), the parallel energetic behavior of SSB-ssDNA and relaxing ssDNA suggests that SSB eliminates the hysteresis seen in ssDNA alone that is introduced by secondary structure formation (*Zhang et al., 2001*) and SSB allows the ssDNA within the complex to behave as though long range secondary structure were absent. The addition of SSB resulted in an additional 3100 ($\pm$550) $k_BT$ (represented as $\Delta\Delta E$) at salt concentrations from 25 to 250 mM NaOAc relative to relaxing ssDNA (*Figure 5—figure supplement 2*, open red circles, dashed line). Only at the highest salt concentration that we measured, 750 mM NaOAc, did we measure a substantial reduction in the $\Delta\Delta E$, consistent with our interpretation that SSB dissociates at these high salt concentrations, but not measurably below 250 mM NaOAc. While the total energy contribution of SSB binding is nearly constant (i.e. isoenergetic) within error, with respect to increasing salt concentrations (up to ~250 mM NaOAc), the contribution of each individual SSB tetramer should (and does) vary due to the transition in the site size across the range of salt conditions. This is—in part—reflected in the complex relationship that increasing salt has on the intrinsic flexibility of the ssDNA (reflected in the 'relaxing' curve, *Figure 5A*, open black circles) vs the folding contributions of secondary structure (reflected in the 'pulling' curve, *Figure 5A*, closed black circles). Because SSB eliminates hysteresis, and we interpret this as complete removal of hysteretic secondary structure, we reason that the relevant comparison (i.e. $\Delta\Delta E$ calculation) is between the +SSB curve (*Figure 5*, red symbols) and 'relaxing' ssDNA (*Figure 5*, open black symbols) (*see Discussion*). Accounting for the change in site size as the concentration of salt increases, the energy of SSB binding to ssDNA in our measurement corresponds to ~11 ($\pm$2) $k_BT$ per SSB tetramer (assuming ~200–320 SSB tetramers per ssDNA molecule), which is in reasonable agreement with the binding energy previously reported (*Zhou et al., 2011*; *Suksombat et al., 2015*). These calculations are limited owing to our inability to actually count the number of SSB tetramers bound to each ssDNA molecule in the magnetic tweezer experiment.

## Binding of RecO and RecOR to SSB-coated ssDNA: implications for homology-dependent annealing and RecA nucleoprotein filament formation

We next asked whether the intramolecular condensation could also be induced by addition of a protein that interacts with SSB and that is presumed to alter its interactions with ssDNA. Therefore,

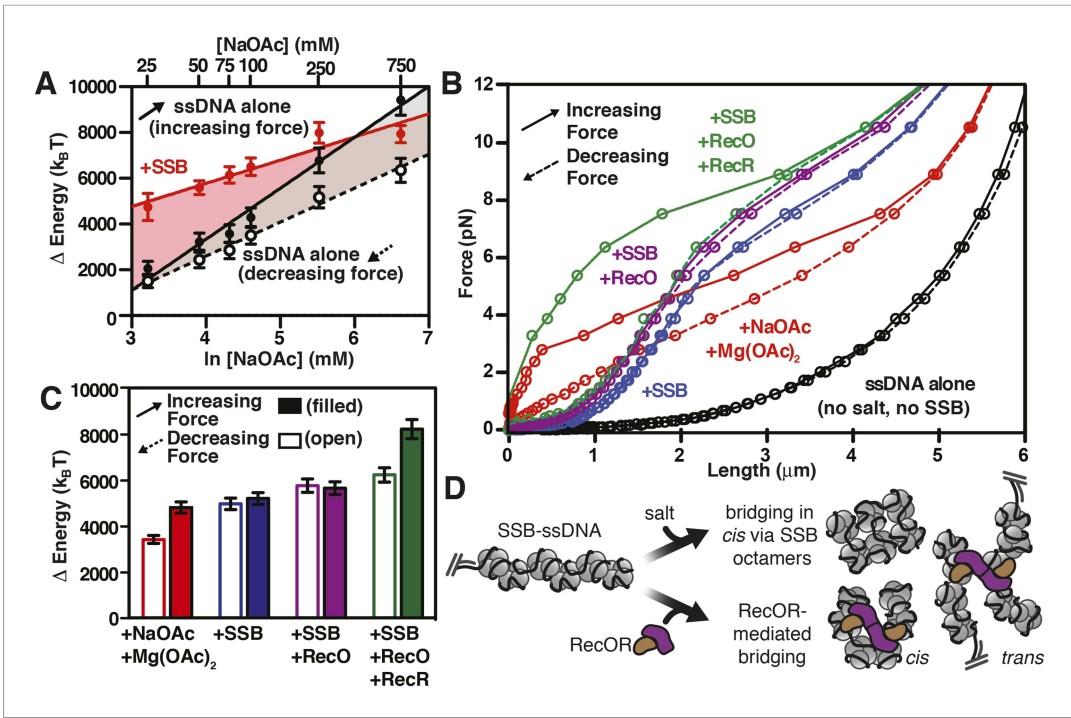

**Figure 5**. RecO and RecOR alter SSB-ssDNA wrapping to induce nucleoprotein fiber condensation (**A**) The work (i.e. ΔE) stored in the ssDNA or SSB-coated ssDNA molecules was determined from the area under the curves from the data in *Figure 4C,D*, as shown in *Figures 5—figure supplements 1, 2*, and plotted as a function of the natural logarithm (bottom x-axis) of NaOAc concentration (top x-axis) for ssDNA (black-filled circles, increasing force; black open circles, decreasing force) and for SSB-coated ssDNA (red filled circles). The lines are linear fits where the slope, $\delta k_B T/\delta \ln[NaOAc]$, is 1000 (±200) for SSB (red line), 1500 (±200) for ssDNA when decreasing force (black dashed line), and 2200 (±150) for ssDNA when increasing force (black solid line). (**B**) The force–extension relationship of a single molecule of ssDNA was measured in the absence of salt (black), then again after sequentially adding each of the following: 100 mM NaOAc and 1 mM $Mg(OAc)_2$ (red), 200 nM SSB (blue), 100 nM RecO (purple), and 1 µM RecR (green). The extension in the presence of RecO results in approximately a 10% condensation at each force measured. In the presence of both RecO and RecR, significant hysteresis is observed (compare solid and dashed green lines). (**C**) The change in energy was determined by integrating the area under the curves in panel B relative to the ssDNA alone curve (black), and are plotted for both increasing (filled bars) and decreasing force (open bars). (**D**) A cartoon depicting our model for salt-induced intramolecular bridging mediated in cis by oligomers of SSB (either tetramers or octamers). RecOR may mediate bridging either in cis, along the same molecule of SSB-coated ssDNA, or in trans to promote annealing of complementary ssDNA.

The following figure supplements are available for figure 5:

**Figure supplement 1**. Diagram of the area under the force-extension curves used to calculate the changes in energy in the absence and presence of SSB.

**Figure supplement 2**. Difference in the change in energy contributed from SSB binding at increasing salt concentrations.

single-molecule force spectroscopy experiments were also performed in the presence of RecO, with and without RecR (*Figure 5B*). For naked ssDNA, in the absence of NaOAc and $Mg(OAc)_2$, the force-extension curves overlap when the applied force is either increased or decreased (i.e. there is no hysteresis) (*Figure 5B*, black curves); however, when 100 mM NaOAc and 1 mM $Mg(OAc)_2$ were added to mimic physiological conditions, significant hysteresis was observed (red curve, *Figure 5B*) due to the salt-induced stabilization of DNA secondary structure (*Smith et al., 1996*). As documented above, this hysteresis in the presence of 100 mM NaOAc and 1 mM $Mg(OAc)_2$ was also completely eliminated upon the addition of SSB (blue curve, *Figure 5B*). When RecO was added in a

stoichiometric ratio to SSB, the nucleoprotein fiber shortened ~10% relative to SSB-alone across the entire force spectrum, and hysteresis was not evident (purple lines, *Figure 5B*). The absence of hysteresis in the presence of RecO is interesting because RecO is known to facilitate annealing of complementary ssDNA in the presence of SSB (*Kantake et al., 2002*), suggesting that RecO functions primarily in trans to promote annealing of complimentary strands, but not in cis to permit formation of non-specific secondary structures.

RecR forms oligomers (dimers/tetramers) (*Kim et al., 2012*) and interacts with RecO (*Umezu and Kolodner, 1994*), but does not interact with SSB, ssDNA or dsDNA (*Radzimanowski et al., 2013*). It also negatively regulates RecO-dependent annealing (*Kantake et al., 2002*) but promotes SSB-ssDNA remodeling in such a way that enhances both the nucleation and growth of RecA filaments on SSB-ssDNA (*Bell et al., 2012*). When we added RecR in the presence of both SSB and RecO, we observed severe hysteresis in the force-extension curves (*Figure 5B*, compare solid and dashed green lines). When the force was decreasing (*Figure 5B*, green dashed line), the curve was essentially indistinguishable from the experiment containing RecO and SSB (*Figure 5B*, purple lines), but when the force was increasing, significantly more force was required to extend the molecule (*Figure 5B*, green solid line). This hysteresis suggests that RecR bridges distant contacts, either through protein–protein interactions or by facilitating intramolecular annealing in cis of regions of micro-homology, and that these contacts are disrupted as the fiber is stretched by the magnetic tweezers.

Similar to the analysis that we showed in *Figure 5—figure supplement 1* and *Figure 5A*, in *Figure 5C* we show the integrated area under the force extension curves to determine the change in energy due to the successive and additive inclusion of buffer containing salt (100 mM NaOAc and 1 mM $Mg(OAc)_2$ (red); SSB (blue); RecO (purple); and RecOR (green). The bars in the plot represent the work (expressed as Δ Energy) done while increasing force (i.e. pulling; filled bars) or decreasing force (i.e. relaxing; open bars) during the experiment. As expected from our previous experiments, the addition of salt induced hysteresis in the force-extension curve, wherein formation of intramolecular secondary structure accounted for approximately 1400 $k_BT$. The addition of SSB was sufficient to eliminate this hysteresis in the presence or absence of RecO. The energy contribution of RecO to the SSB-ssDNA energy was modest, accounting for ~250 $k_BT$ (or ~1.2 $k_BT$ per SSB tetramer). The addition of RecR—so that SSB, RecO, and RecR were all present—again, contributed a modest amount of energy when the complex was relaxing (an additional ~220 $k_BT$ or ~1 $k_BT$ per SSB tetramer); however, when RecOR was bound to the SSB-coated ssDNA, hysteresis returned to the system, which we interpret as a result of long-range, protein–protein interactions bridged by RecOR bound to SSB-ssDNA. The energy contribution from RecOR-mediated bridging was ~2000 $k_BT$ (or ~10 $k_BT$ per SSB tetramer). Interestingly, this energy contribution is essentially the same as the energetic contribution from SSB binding to ssDNA alone (~9–13 $k_BT$ per tetramer, see previous section), indicating that RecOR bridges distant SSB tetramers and those tetramers must be disrupted as the molecule is stretched by increasing force. We conclude that when incubated together, RecOR forms a complex that binds to SSB-bound to ssDNA, and in doing so, both modulates the wrapped-state, or binding-mode, of SSB in order to induce macroscopic changes and serves as a scaffold to bridge distant sites along the SSB-ssDNA nucleoprotein fiber.

## Discussion

In this work, we comparatively analyzed how the length of SSB-coated ssDNA is modulated by salt and protein binding partners, correlating measurements of ensemble equilibrium binding, direct visualization of single molecules, and single-molecule force spectroscopy. These results provide an extensive physical description of the polymer dynamics of SSB-coated ssDNA and reveal a previously unrealized property of SSB-ssDNA complexes to interact with distant intramolecular sites, which is manifest as condensation of single nucleoprotein fibers. This macroscopic condensation could occur either through the association of stable octamers or through ssDNA by the direct binding of distant sites to a single tetramer. The extent of condensation, however, is greatest under conditions where ssDNA is fully wrapped around the protein in the $SSB_{65}$ binding mode, making the former possibility more likely. In our force-spectroscopy experiments, the addition of SSB (or SSB and RecO) eliminated the salt-induced hysteresis caused by formation of DNA secondary structure through intramolecular annealing in cis. When we compare the energetic contribution of SSB binding to ssDNA, relative to the salt-induced effects on the ssDNA polymer energetics (excluding the contributions from long range

secondary structure by comparing the 'relaxing' ssDNA force-extension curves), the change in energy due to SSB binding (when summed over a single molecule of ssDNA) is nearly isoenergetic across a large salt concentration range (*Figure 5A*, compare the slope for SSB, red-filled circles, to relaxing ssDNA alone, open black circles, and *Figure 5—figure supplement 2*). This is both interesting and unexpected as it implies a homeostasis with regard to the energetics of SSB, relative to ssDNA, at physiological salt concentrations, making the absolute intracellular salt concentration nearly unimportant relative to other cellular processes working on or with the SSB-ssDNA complex. We propose that this relatively constant energy contribution across the ssDNA is made possible by a net change in the microscopic binding modes of SSB in such a way that all (or most) ssDNA is effectively coated by SSB, albeit with each SSB tetramer engaged in a different number of nucleotides and a different number of SSB tetramers engaged at each salt concentration. In this way, the sum of the energy changes (see *Suksombat et al., 2015*) for each tetramer is balanced across a wide range of physiological conditions.

Importantly, this 'isoenergetic landscape' is independent of salt only when we compare the SSB-ssDNA curves with the relaxing conditions for ssDNA alone (*Figure 5A* and *Figure 5—figure supplement 2*), where secondary structure considerations are experimentally removed. From the perspective of displacing SSB from ssDNA by another protein, this is the biologically relevant comparison because the absence of hysteresis in the SSB-ssDNA curves shows that there is no long range secondary structure in the SSB-ssDNA complexes. When we calculate the ΔΔE relative to the 'pulling' condition with ssDNA alone where the contributions of secondary structure are clearly evident (*Figure 5A*, solid black circles and *Figure 5—figure supplement 2*, solid red circles, solid red line), the ΔΔE decreases until it becomes negative (and therefore unfavorable) at approximately 400 mM NaOAc, corresponding to a concentration of salt where secondary structure may become more stable than the SSB–ssDNA interaction (consistent with our observations in *Figure 1—figure supplement 1*). Therefore, this net energetic accounting reflects what would be apparent in a traditional ensemble measurement, where SSB-binding and DNA secondary structure formation are in competitive equilibrium, and reconciles our observations with previous work that has interrogated the salt-dependence of SSB-ssDNA interactions where the standard state for free energy calculations is the free ssDNA that will form secondary structure.

Previous AFM imaging of SSB bound to M13mp7 ssDNA (7249 nts) measured contour lengths of SSB-ssDNA of ~920 nm in the low salt, 35-nucleotide binding mode, and 560 nm in the high salt, 65-nucleotide binding mode; this change represents a 1.6-fold increase upon going from high to low salt, or a decrease of 60% upon going from low to high salt (*Hamon et al., 2007*). Since denatured λ-phage DNA is 6.6-fold longer than M13mp7 ssDNA, we expected our SSB-ssDNA would be approximately 6.1 μm long in low salt and compacted by 1.6-fold to approximately 3.7 μm long in high salt (up to ~300 mM Na⁺); however, we observed an approximately fourfold compaction over this range. A substantially greater degree of compaction of the complexes was measured upon further increase in salt concentration – from 6.5 μm in the absence of salt to 0.5 μm at 750 mM NaOAc – which is a 13-fold compaction. If the site size was the dominant factor determining the condensation state of SSB-coated ssDNA, then the maximum change in length should be approximately twofold, yet we measure a ~13-fold decrease in length. We propose that an explanation derives from the fact that SSB does not simply bind to ssDNA as an array of globular units, but instead exhibits limited cooperativity between tetramers, at higher salt concentrations, where it also forms octamers (*Lohman and Bujalowski, 1988*; *Bujalowski and Lohman, 1989a, b*) which might serve as an intramolecular bridges between distant sites along the ssDNA resulting in the condensation that we observe (*Chrysogelos and Griffith, 1982*; *Griffith et al., 1984*; *Hamon et al., 2007*). Although the molecular nature of the condensing species is unknown, we also note that the population of the various SSB-binding modes depends not only on intracellular solution conditions, but also on the SSB concentration itself due to the established effects of 'competition' between DNA-binding modes with different site sizes, affinities, and cooperativities (*Schwarz and Stankowski, 1979*; *Bujalowski et al., 1988*); therefore, with SSB in excess over ssDNA – which is the cellular situation – multiple modes coexist and may serve to keep the amounts of bound SSB nearly constant. Finally, although the high degree of salt-induced compaction of SSB-ssDNA that we observed was surprising based on previous AFM and EM studies, our results are in good qualitative agreement with early, largely unexplained, ultracentrifugation experiments performed with SSB-saturated M13-phage ssDNA (*Schaper et al., 1991*): our changes in length with salt concentration map exactly on the reported changes in sedimentation coefficients for the SSB-M13 ssDNA complexes.

Importantly, the force-extension measurements of unlabeled, wild-type SSB recapitulates the relative condensation of SSB$^f$-coated ssDNA observed in the direct visualization experiments, indicating that the observed condensation of the fluorescently modified nucleoprotein complex in *Figure 1C* and *Figure 3C–G* is not a consequence of using modified SSB, SSB$^{AF488}$, or SSB$^f$, but rather is due to an intrinsic, salt-induced conformational change resulting in intramolecular re-organization along the nucleoprotein fiber. The absence of hysteresis in our force-extension curves in the presence of SSB is strong evidence that the intramolecular condensation that we observe is not due to the formation of distant ssDNA–ssDNA contacts, but rather is driven by the microscopic reorganization of SSB along the ssDNA, which in turn contributes to a macroscopic folding of the molecule. Several structural polymer models might explain the condensed molecules that we observe, including the formation of solenoid or fractal structures; however, several interesting properties of SSB support the idea that intramolecular folding or condensation is protein-mediated. First, dimerization of SSB tetramers (i.e. octamer formation) is a well-established phenomenon at increasing but still physiologically relevant intracellular salt concentrations (*Chrysogelos and Griffith, 1982*; *Bujalowski and Lohman, 1987*), which can linearly vary from 0.23 to 0.93 molal K$^+$ ion and from 0.03 to 0.26 molal glutamate ion in response to changes in the osmolality of the growth medium (*Richey et al., 1987*) *see also*, (*Epstein and Schultz, 1965*) (*Epstein and Schultz, 1966*). Second, SSB exhibits long-range, intersegmental transfer (*Lee et al., 2014*), the latter of which likely proceeds through direct, tetramer transfer within a ternary intermediate comprising SSB and two ssDNA molecules without proceeding through a free-protein intermediate (*Kozlov and Lohman, 2002a, b*). This property of SSB is possible owing to multiple, high affinity binding sites distributed around the tetramer, or as the case may be in the high-salt mode, octamers. This phenomenon has primarily been described as a transient intermediate during rapid, stopped flow kinetics; however, we believe that the intramolecular condensation that we observe here provides evidence that the sum of these transient interactions across the SSB-ssDNA fiber might explain how many SSB tetramers can simultaneously engage in distant ssDNA sites to contribute to an intramolecular, folded polymer that is highly dynamic and 'fluid', undergoing constant, steady–steady state protein turnover and diffusion at equilibrium. This is corroborated by the rapid exchange of SSB protein (*Figure 1F*) and the absence of net protein loss under steady-state conditions (*Figure 1D–E* and *Figure 1—figure supplement 1*), but the rapid dissociation in the absence of free protein during salt-jump experiments (*Figure 2—figure supplement 1*), as well as the ability of SSB labeled with different fluorophores to exchange and form mixed complexes (JCB, unpublished observations), a phenomenon also demonstrated for eukaryotic RPA (*Gibb et al., 2014*).

Similar to nucleosomes, SSB binds and wraps a DNA polymer around itself (albeit ssDNA instead of dsDNA), interacts with dozens of proteins via a short acidic peptide tail, and exhibits complex cooperative and anti-cooperative behavior that is modulated by salt concentration. Our observation that SSB-ssDNA is macroscopically organized and regulated through microscopic interactions is surprisingly similar to the most basic organization of eukaryotic chromatin and highlights the important role of SSB, not simply as a kinetic trap for ssDNA, but as an organizational and regulatory scaffold during DNA metabolism (*Shereda et al., 2008*; *Sun et al., 2015*). This organizational and regulatory role is likely controlled by the acidic, intrinsically disordered C-terminal tail of SSB, which is required for cooperative binding of SSB to ssDNA (*Kozlov et al., 2015*). In the absence of interaction partners or ssDNA, this unstructured peptide tail interacts with the subunits within the SSB tetramer (*Kozlov et al., 2010a*). Recent studies have identified a cadre of proteins, including the χ subunit of DNA polymerase III, PriA, PriB, RecG, RNaseHI, Exonuclease I, and RecO, that bind to the C-terminal tail of SSB and either remodel the SSB-ssDNA complex or regulate enzymatic ssDNA metabolism (*Cadman and McGlynn, 2004*; *Shereda et al., 2007, 2008*; *Lu and Keck, 2008*; *Kozlov et al., 2010a, 2010b, 2015*; *Wessel et al., 2013*; *Bhattacharyya et al., 2014*; *Petzold et al., 2015*; *Sun et al., 2015*). Extrapolating from our observations in this work, these many proteins may regulate access to ssDNA by binding to SSB and altering either compaction/de-compaction most likely by perturbing the microscopic binding state of SSB.

Indeed, we demonstrate that this condensation occurs not only with increasing osmolality, but also by the addition of RecO, which binds directly to SSB, in the absence and presence of RecR (*Ryzhikov et al., 2011*). As the C-terminal tail of SSB interacts with more than a dozen proteins involved in DNA replication, recombination, and repair (*Shereda et al., 2008*), our observation supports and expands upon the idea that these proteins might modulate the macroscopic condensation state of the SSB-ssDNA fiber by microscopically altering the binding mode, and therefore either grant or restrict access

to the ssDNA. In the case of RecO and RecOR, altering this macromolecular state by bridging distant sites could reduce the three-dimensional space required to facilitate homology-dependent annealing of ssDNA (*Berg et al., 1981*; *Forget and Kowalczykowski, 2012*), which occurs by a second order kinetic process (*Berg et al., 1981*; *Kantake et al., 2002*; *Wu et al., 2006*; *Bell et al., 2012*). Similarly, RecO has been shown to slow the rate of one-dimensional diffusion, or sliding, of SSB on ssDNA (*Zhou et al., 2011*), which is consistent with the small energy contribution of RecO and RecOR binding we observe here, where RecO contributes an additional ~1–2 $k_BT$ per SSB tetramer. By slowing the rate of diffusion of SSB on ssDNA, RecO might facilitate both annealing and RecA nucleation by increasing the lifetime of transiently exposed ssDNA. The long range interactions induced by RecR could also be due to phasing of SSB via its interaction with RecO, creating microscopic gaps on the ssDNA that allow distant sites with micro-homology to anneal, forming intramolecular secondary structure (*Figure 5D*). In its biological context, these microscopic gaps could also facilitate the nucleation of RecA filaments during homologous recombination by exposing short segments of ssDNA between SSB tetramers, either by inducing a conformational change that physically disrupts the SSB-ssDNA complex to create gaps or by increasing the lifetime of transiently exposed ssDNA created during SSB sliding along the ssDNA (*Bell et al., 2012*).

## Materials and methods

### Direct visualization of SSB-coated ssDNA complexes

Fluorescent SSB was generated by conjugating either Alexa Fluor 488 maleimide or fluorescein-5-maleimide (Life Technologies) to SSB$^{G26C}$ as previously described (*Dillingham et al., 2008*). Bacteriophage λ DNA (1.5 nM molecules) was biotinylated by incorporating biotin-dGTP (50 μM) at the 3′-ends of DNA using T7 Polymerase (10 Units) in NEB Buffer 2 (10 mM TrisHCl (pH 7.9), 50 mM NaCl, 10 mM MgCl$_2$, and 1 mM dithiothreitol (DTT)) in the presence of 50 μM dATP, dCTP, and dTTP for 15 min at 12°C. The reaction was terminated by the addition of 20 mM EDTA and incubated at 75°C for 10 min. The biotinylated dsDNA was purified from unincorporated biotin-dGTP using an S-400 spin column equilibrated with 20 mM TrisHCl (pH 7.5) and 0.1 mM EDTA. This biotinylated dsDNA was then diluted to 250 pM (molecules) in 10 μl of 0.5 M NaOH for 10 min at 37°C and subsequently diluted into 400 μl of buffer containing 20 mM TrisOAc (pH 8.0), 20% sucrose, 50 mM DTT, and 200 nM of the indicated SSB-derived biosensor. The final concentration of ssDNA was 12.5 pM molecules or 600 nM nucleotides. The ssDNA-nucleoprotein complexes were then injected into a flow cell and tethered to the surface of a coverslip. Flow cells (4 mm × 0.4 mm × 0.07 mm) were assembled using a glass slide, a coverslip, and double-sided tape (3M Adhesive Transfer Tape 9437). Ports were drilled into the glass microscope slide, and flow was controlled using a motor-driven syringe pump (*Amitani et al., 2010*; *Forget and Kowalczykowski, 2010*, *2012*). The surface of the coverslip was cleaned by the subsequent injection of 1 M NaOH for 10 min, rinsed with water and equilibrated in buffer containing 20 mM TrisOAc (pH 8.0), 20% sucrose and 50 mM DTT. The surface was then functionalized by injecting the above buffer containing 2 mg/ml biotin-BSA (Pierce) and incubated for 10 min, rinsed with buffer, equilibrated with 0.2 mg/ml streptavidin (Promega) for 10 min and then blocked with 1.5 mg/ml Roche Blocking Reagent (Roche) for 10 min.

For imaging, the nucleoprotein complexes were allowed to incubate in the flow cell in the absence of flow for approximately 5–15 min until a sufficiently desired density of molecules were tethered to the surface, then visualized using TIRF microscopy while extended by flow at volumetric flow rate of 4000 μl/hr. Unless otherwise noted, imaging was performed in 20 mM TrisOAc (pH 8.0), 50 mM DTT, 20% sucrose, and the indicated concentration of NaOAc or Mg(OAc)$_2$. Unless otherwise indicated (as in *Figure 2*), the concentration of SSB was 200 nM (monomers).

### Equilibrium fluorescent binding assays

Titrations to monitor the binding of SSB$^f$ to ssDNA were performed by monitoring the fluorescence enhancement at 25°C, using a Shimadzu fluorescence spectrophotometer set at an excitation wavelength of 495 nm and an emission wavelength of 520 nm. Excitation and emission slits were set to a bandwidth of 3 and 10 nm, respectively. The concentration of SSB$^f$ was 100 nM (tetramer). Titrations were performed in 20 mM TrisOAc (pH 8.0), 1 mM DTT, and the indicated concentration of salt. The

fluorescence values were corrected for dilution and normalized to the fold increase in fluorescence (fluorescence intensity of SSB[f] plus ssDNA divided by the SSB[f] fluorescence in the absence of ssDNA). The site size of SSB was determined by fitting the data to a two-segment line, where the x- and y-intercepts of the first segment and the slope of the second segment were constrained to zero. The x-intercept between the segments was taken to be the stoichiometric breakpoint of the titration. Data fitting was performed using GraphPad Prism version 5.0d. All equilibrium titrations were performed in triplicate and report the mean and standard deviation from each experiment.

## Force spectroscopy with magnetic tweezers

The multiple-DIG and multiple-biotin labeled 2-kb DNA handles were prepared using pUC19, which was linearized by HindIII, as the template. The primer sequences for multi-DIG labeled 2-kb DNA handle were primer-1 (5′-GTT GTG GGC CCG GCG TAA TCA TGG TCA TAG CTG-3′) and primer-2 (5′-CAA CAT TTC CGT GTC GCC CTT ATT CCC-3′). Primer-1 creates a restriction site for ApaI (underlined); dsDNA length is 2036 bp after PCR and 2026 bp after Apa1 digestion. PCR was performed in the presence of 0.2 mM dATP, dGTP, dCTP, 0.18 mM dTTP, and 0.02 mM DIG-11-dUTP (Roche) using Taq DNA polymerase (NEB). For the DNA handle containing biotin, primer-3 (5′-GTT GTG CTA GCG GCG TAA TCA TGG TCA TAG CTG-3′) was used instead of primer-1. Primer 3 creates a restriction site for NheI (underlined); dsDNA length is 2036 bp after PCR and 2030 bp after NheI digestion. The PCR was performed in the presence of 0.2 mM dATP, dCTP, dTTP, 0.18 mM dGTP, and 0.02 mM biotin-11-dGTP (Perkin–Elmer). The ~13.5-kb DNA was prepared using lambda DNA as the template using primer 4 (5′-GTT GTG GGC CCA CCA CCT CAA AGG GTG ACA G-3′) and primer 5 (5′-GTT GTG CTA GCA CGG TGG AAA CGA TAC TTG C-3′) to produce dsDNA 13,572 bp in length. These primers create restriction sites for ApaI and NheI (underlined), respectively. PCR was performed using Expand 20kbPLUS PCR system (Roche). The PCR products were digested with appropriate restriction enzymes to yield dsDNA 13,552 bp in length and purified using a Qiagen PCR purification kit (Qiagen). All three pieces of DNA were ligated in a single step. Flow cells were assembled by sandwiching double-sided tape (3M Adhesive Transfer Tape 9437) from which a rectangular channel had been cut with a precision controlled razor blade printer (Craft Robo CC200-20, Graphtec) between a Mylar sheet (0.002′′, McMaster) and a coverslip (No. 1;113 Corning). Flow cells were washed with water, phosphate buffered saline (PBS; Gibco #10010; 1 mM $KH_2PO_4$, 3 mM $Na_2HPO_4$, and 155 mM NaCl, pH 7.4) and then coated with 0.2 mg/ml anti-digoxigenin (Roche) in PBS by incubating at 37°C overnight. Unbound anti-digoxigenin was rinsed with PBS. The surface was blocked for at least 2 hr at 37°C with a solution containing 10 mg/ml BSA (Sigma), 3.3 mg/ml poly-L-glutamic acid (Sigma) in 45 mM $NaHCO_3$ (pH 8.1) and 50 mM DTT. The blocking agent was rinsed from the flow cell using single molecule buffer (SMB) containing 20 mM TrisOAc (pH 8.0) and 50 mM DTT, and then blocked again using 1.5 mg/ml Roche blocking reagent (RBR) dissolved in SMB plus 1 M NaOAc for 30 min, followed by a successive incubation of 1.5 mg/ml RBR in SMB (no salt) for an additional 30 min. The flow cell was mounted onto a PicoTwist microscope (PicoTwist, Paris, France) so that the Mylar is under tension. The position of the magnets was carefully adjusted so that the distance between the magnets and the flow cell surface was accurate and calibrated according to the manufacturer's specifications. The DNA substrate was incubated in 0.5 M NaOH to denature the dsDNA to ssDNA, which was then attached to magnetic beads (1 μm MyOne C1 Dynal) by mixing biotinylated ssDNA and the beads in ~5:1 molar ratio in SMB and incubating on a slow rotator for 15 min at room temperature. The ssDNA-bead mixture was added to the flow cell. After 10-min incubation at 25°C, untethered beads were eliminated by extensive washing with SMB. Changing the position of the magnets controlled the force. Flow was driven by gravity. The flow cell was sealed by switching off the inlet and outlet valves after buffer exchange, the magnets were moved to their destination position, and data collection began.

After each force-extension curve was obtained, the buffer in the flow cell was exchanged to increase the concentration of salt, and the experiment was repeated. Experiments containing SSB were similarly obtained, where the force-extension curve of ssDNA alone was obtained in the absence of salt, then plus 200 nM SSB, and finally incrementally increasing the salt concentration but maintaining a constant concentration of SSB. All experiments were performed in SMB plus the indicated concentration of salt at 25°C. RecO and RecR were purified as previously described (*Kantake et al., 2002*). Experiments were performed by sequentially adding each of the following components to SMB and injecting the solution into the flow cell after ssDNA-bead complexes were tethered to the surface: 100 mM NaOAc and 1 mM

Mg(OAc)$_2$, 200 nM SSB, 100 nM RecO, and 1 µM RecR. Each additional component was added to the previous buffer, and the molecules were allowed to equilibrate for 5–10 min before the experiment.

The change in energy was measured using the 'Area Under Curve' (AUC) function in GraphPad Prism (v5.0d) for each molecule at either increasing or decreasing force and subtracting the AUC for ssDNA alone in the absence of salt. At the highest salt concentration measured (750 mM NaOAc), 17.5 pN was sufficient to completely dissociate SSB from ssDNA, as ascertained by the convergence of the force-extension curve with ssDNA alone. At all other salt concentrations, the maximum force applied was 10.5 pN, and the curve was completed by extrapolating between the 10.5 pN and 17.5 pN in order to complete the curves to calculate the integrated area. By comparing the measurements in the 750 mM NaOAc data set with a linear extrapolation from 10.5 pN to 17.5 pN, we calculate that this analysis contributes no more than 370 pN•nm to the error in our measurement, which is reflected in the error bars in Figures 6B and 6D. The AUC was converted from energy units of pN•nm to $k_BT$ using the conversion, 1 $k_BT \sim$ 4.1 pN•nm (*Nelson, 2004*).

## Acknowledgements

We are grateful to members of the laboratory for their comments on this work. JCB. was funded by the UCD-NIH Pre-doctoral Training Program in Molecular & Cellular Biology (T32 GM007377) and SCK was supported by the National Institutes of Health (GM-62653 and GM-64745).

## Additional information

### Competing interests

SCK: Reviewing editor, *eLife*. The other authors declare that no competing interests exist.

### Funding

| Funder | Grant reference | Author |
| --- | --- | --- |
| National Institute of General Medical Sciences (NIGMS) | GM-62653 | Stephen C Kowalczykowski |
| National Institute of General Medical Sciences (NIGMS) | GM-64745 | Stephen C Kowalczykowski |
| National Institute of General Medical Sciences (NIGMS) | T32 GM007377 | Jason C Bell |

The funder had no role in study design, data collection and interpretation, or the decision to submit the work for publication.

### Author contributions

JCB, Conception and design, Acquisition of data, Analysis and interpretation of data, Drafting or revising the article; BL, Acquisition of data, Analysis and interpretation of data; SCK, Conception and design, Analysis and interpretation of data, Drafting or revising the article

### Author ORCIDs

Jason C Bell, http://orcid.org/0000-0001-5480-7975
Bian Liu, http://orcid.org/0000-0001-8806-820X
Stephen C Kowalczykowski, http://orcid.org/0000-0002-9127-3949

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
