## [Decision Letter]

Thank you for submitting your work entitled “Imaging and energetics of single SSB-ssDNA molecules reveal intramolecular condensation and insight into RecOR function” for peer review at eLife. Your submission has been evaluated by James Manley (Senior editor) and three reviewers, one of whom is a member of our Board of Reviewing Editors.

The reviewers have discussed the reviews with one another and the Reviewing editor has drafted this decision to help you prepare a revised submission. Bell et al. examine the properties of single molecules of SSB-coated ssDNA using TIRF microscopy and force spectroscopy (magnetic tweezers). The first set of experiments use tethered single-stranded Lambda DNA coated with fluorescently labeled SSB in a flow cell to image the condensation/length of the molecules in response to salt titrations. The degree of condensation is greater than what was expected based on simple binding mode transitions, and it is suggested that this is due to long-range intramolecular interactions. The second set of experiments use force spectroscopy to generate force-extension curves of Lambda ssDNA coated with unmodified SSB in the absence and presence of RecO/RecOR. The results from this indicate that addition of RecO alone further shortened the nucleoprotein fiber and the addition of RecOR induced significant hysteresis suggesting RecR is bridging distant contacts within the nucleoprotein fiber. The authors conclude that RecOR binding to SSB-ssDNA modulates the binding mode of SSB and serves as a scaffold to condense distant sites on the nucleoprotein fiber.

The data in this manuscript are well presented and the novel insights into the binding mode transitions that the *E. coli* SSB can assume are intriguing. The idea that the overall condensation state of the SSB/single strand DNA filament may be able to change as a result of higher order interactions with other proteins has been around for some time. However, the work presented here for the recombination proteins seems rather definitive and important.

In summary, this is certainly an interesting study and the SSB-induced ssDNA compaction observed at high salt may be larger than expected based on the known SSB-ssDNA binding mode transitions in the literature. However, the extent of any additional compaction of ssDNA by SSB beyond that expected based on the SSB-ssDNA binding mode transitions is difficult to assess quantitatively. In particular the major problems raised by all three reviewers concerns perhaps the validity of predicting expected compaction of ss lambda DNA by SSB should be based on site sizes measured with polydT (for example see point 4 and 7 below).

The major points are listed below:

1) The manuscript addresses the transitions in binding modes as a function of salt concentrations and the reviewers don't understand how the present data can be reconciled with published similar measurements albeit with different tools that the authors mention. In these previous studies such dramatic compaction was not reported. This was confusing and perhaps a discussion clarifying the anticipated reasons for the differences could be made.

2) Seeing is believing, but given the low spatial resolution of the methods certain structural assumptions need to be made with regard to fiber dimensions and as such compaction calculations are perhaps potentially in error? Is the model put forward that the compaction is due to long range interactions between different SSB tetramers the only one that can explain the data?

3) The analogies to nucleosome condensation with salt changes is intriguing but the physiological relevance of this phenomenon has yet to be established for chromatin. While osmolarity changes in a prokaryote cell should be significant are they in the range used by Bell et al. for *E. Coli* in its natural environment? Perhaps some references here are appropriate.

4) It is well established that *E. coli* SSB can bind single stranded DNA in multiple binding modes that differ in the extent of wrapping of the ssDNA around the tetramer. In experiments with poly(dT), binding modes with occluded site sizes of ∼35, 40, 55 and 65 nucleotides per tetramer have been identified. The relative stability of these modes can be modulated in vitro by changing the salt concentration. As the salt concentration is increased, the mode of binding shifts from the 35 mode to the higher site size modes, resulting in progressive compaction of the ssDNA. These different binding modes also display very different cooperative behavior. The binding mode transitions have also been demonstrated by sedimentation, EM and AFM measurements. The previously reported occluded site size measurements as a function of salt type and concentration have generally been performed using poly(dT) in order to eliminate complications due to secondary structure (hairpins, etc.) that will form in natural ssDNA, such as M13 or lambda DNA that is used in the current study. In the original report of salt-induced SSB binding mode transitons (38) the apparent site sizes measured using poly(dT) were compared with measurements using ss M13 phage DNA. These showed that the apparent site size is severely overestimated when M13 ssDNA is used and that the discrepancy increases as the salt concentration increases (e.g., 65 nt for pol(dT) vs. 77 nt for M13 DNA at 0.3 M NaCl). This difference will be greater at higher salt concentrations. This difference is due to the fact that SSB, especially at high salt concentrations is unable to remove all of the secondary structure from natural ssDNA. This is partly because SSB does not bind with high cooperativity in the 55 or 65 modes. Therefore, the uncertainty in predicting what the expected compaction of ss lambda DNA based on site sizes measured on polydT makes the main conclusion difficult to assess. The authors need to address this before the manuscript can be published.

5) As the authors mention, the extent of ssDNA compaction observed by both EM (Griffith) and AFM (Harmon et al.) studies is consistent with the extent of compaction expected from the SSB binding mode transitions. Compaction as measured by sedimentation studies ([12] JBC 263, 4629) show that compaction tracks with occluded site size, when both measurements are performed with poly(dT). It seems that the additional ssDNA compaction suggested in the current study should be observable also by these techniques, but none has been reported. Why?

6) Bell et al. use minimum and maximum dimensions of SSB of 6 nm and 8 nm for the calculation of the “expected” extent of ssDNA compaction, but do not explain how they arrived at these estimates. Reviewers agree with the 8 nm estimate, which is from the tips of the L45 loops in the crystal structure. However, the other dimension from the crystal structure can be as small of 4.5 nm. When the calculation is done using 4.5 nm, the predicted length lower limit is lowered by 25%, which encompasses all of the data points except for the two at the highest salt concentrations. These two points are subject to the caveats mentioned elsewhere in the review.

7) Bell et al. measured the SSB-induced ssDNA compaction using mixed sequence ssDNA (lambda), whereas the site size measurements are made using poly(dT). Hence, some of the “unexpected” additional compaction is certainly due to unmelted secondary structure in the lambda DNA. How much is not clear. The difference in site sizes measured using poly(dT) vs. natural ssDNA (mixed sequence) will be greater at the higher salt concentrations (> 0.3 M), consistent with the authors reporting the largest “additional unexpected” compaction at these higher salt concentrations.

8) Potential SSB dissociation. The authors initially state that the amount of SSB bound to the ssDNA during the collapse transition is “essentially constant”, attributing the loss of SSB fluorescence to photobleaching. Yet they also show that at the very high salt concentrations (> 750 mM), some of the SSB does dissociate, although this is not considered to be a major factor. The question is how much dissociates and is it reasonable to dismiss this? SSB does not bind as tightly to natural ssDNA as it does to poly(dT), hence this is likely not inconsequential. Please explain.

9) Could the constant fluorescence intensity observed upon shortening of the molecule (Figure 1) be the result of altering the environment of the fluorophore? As indicated, this result is surprising given that increasing the salt should theoretically decrease the number of available binding sites on the ssDNA. Is constant fluorescence observed when no free SSB is included in the flow (Figure 2)? Additionally, we are missing the connection for how this result led to the authors asking whether condensation is reversible (subsection “Single molecules of SSB-coated ssDNA reversibly condense in response to increasing salt concentration”. “We therefore asked”) – clarification on the reasoning as well as a brief mention of possible explanations would be beneficial in this section.

10) Is the salt-induced condensation reversible with free protein and not just in the absence of it (Figure 2)?

11) Clarify how the values plotted in Figure 3 were obtained – was a specific nt/SSB ratio used or were titrations done as in 3A for all of the [NaOAc] in 3B?

12) In the subsection “Force spectroscopy of single molecules of ssDNA and SSB-coated ssDNA reveals a nearly complete relief of hysteresis in SSB-ssDNA unfolding transitions”, second paragraph, it is mentioned that the modest hysteresis at high [NaOAc] is consistent with the interpretation that the decrease in fluorescence enhancement with SSBf at high [NaOAc] is due to partial dissociation. This is the first time that this interpretation and the fact that fluorescence enhancement decreases at high [NaOAc] is mentioned – a brief discussion should be provided earlier within the Results section referring to Figure 3.

13) The figure description for Figure 5 is difficult to follow. In the figure it appears as though purple and gray represent a single area (separate from light purple), however the description seems to be written as if they are all separate.

14) The discussion of nucleosomes and chromatin seems unnecessary - the statement “our observation that SSB...during DNA metabolism” (Discussion, third paragraph) would be sufficient and can be added on to the end of the preceding paragraph.

15) The suggestion in the Discussion that SSB-interacting partners might modulate the condensation state of the SSB-ssDNA fiber by altering the binding mode is supported by work on the interaction between SSB and both PriA and PriC (6, 63). Both proteins have been shown to induce the 35nt binding mode of SSB in a manner dependent upon complex formation.

---

## [Author Response]

*The data in this manuscript are well presented and the novel insights into the binding mode transitions that the* E. coli *SSB can assume are intriguing. The idea that the overall condensation state of the SSB/single strand DNA filament may be able to change as a result of higher order interactions with other proteins has been around for some time. However, the work presented here for the recombination proteins seems rather definitive and important*.

*In summary, this is certainly an interesting study and the SSB-induced ssDNA compaction observed at high salt may be larger than expected based on the known SSB-ssDNA binding mode transitions in the literature. However, the extent of any additional compaction of ssDNA by SSB beyond that expected based on the SSB-ssDNA binding mode transitions is difficult to assess quantitatively. In particular the major problems raised by all three reviewers concerns perhaps the validity of predicting expected compaction of ss lambda DNA by SSB should be based on site sizes measured with polydT (for example see point 4 and 7 below)*.

General response:

First, we'd like to thank the reviewers for their constructive comments, several of which seemed to focus on two major issues:

1) Is it valid to cross reference site size measurements performed with poly(dT) with the single-molecule length measurements, and

2) Does increasing salt concentration cause SSB to dissociate and secondary structure to form, and would this ‘ssDNA collapse’ explain the condensation observed?

We first address these issues generally, and then we will refer back to our general comments, where applicable, as we address the more specific issues point-by-point.

With respect to major issue #1, after considering the reviewers’ comments, we feel that our original Figure 3 was taken more literally than we had intended. The figure panel was meant to illustrate the surprising result that SSB-ssDNA compacted more than we initially anticipated based on the published results in Hamon et al., where they observed a contour length of 920 nm at low salt concentration, where SSB is typically in the “SSB_35_-mode”, and 560 nm in a higher salt concentration, where it is in the “SSB_65_-mode”. This change in length (920 nm vs. 560 nm) represents a 1.6-fold increase upon going from high to low salt, or a decrease of 60% upon going from low to high salt. Since lambda DNA is 6.6-fold longer than ssM13, we anticipated our SSB-ssDNA molecules should be ∼6.1 um in the low salt, SSB_35_-mode and decrease by 60% to ∼3.7 um in the high salt SSB_65_-mode. Both of these DNA molecules are natural, mixed sequences, and we expect their propensity to form secondary structure should be similar and comparable. Though we still refer to these lengths and the typical SSB-binding modes in the text, because we now focus our comparisons to observed native DNA lengths, we have removed the figure plotting our measured site size vs. length, have clarified the different ssDNA molecules used in the individual panels in Figure 3, and have removed the panel (E) showing length predictions.

Our measurements at 250 mM NaOAc should be comparable with the 300 mM NaCl measurements from the Hamon paper, yet our molecules are fully half the expected length (∼2 um measured vs. ∼3.7 um expected). We note that both of these experiments were performed on natural ssDNA, and if anything, we expected our molecules to be longer owing to extension in flow and the absence of spermidine (which they used to adsorb the molecules in the AFM experiment, and which condenses both ssDNA and dsDNA). Our intensity measurements indicate that the amount of fluorescent protein doesn’t change in the salt-jump experiments up to 400 mM NaOAc, where we see begin to see a modest 7% decrease in intensity – this result shows that the SSB has not dissociated at 250 mM NaOAc. At lower salt concentrations, we see no net dissociation of fluorescent SSB when SSB concentrations are held constant. To clarify this important point, we have also included supplemental panels for Figure 1, showing intensity measurements on SSB-ssDNA molecules during the length transitions at multiple salt jumps.

Using a well-established standard ensemble method, we used poly(dT) to determine the intrinsic site size for our fluorescent SSB protein, under our conditions. These measurements allowed us to determine the dominant, intrinsic binding mode for SSB at each concentration of salt used in our single molecule experiments. Poly(dT) is used as the polynucleotide of choice for SSB-binding experiments precisely for the reasons mentioned by one of the reviewers – the absence of secondary structure reduces the complexity of interpreting the site size determined using stoichiometric titrations (or affinity, as the case may be). We used these experiments as verification that our SSB protein is behaving as expected under our conditions, using this method that has been extensively documented by the Lohman lab.

With respect to major issue #2, SSB dissociation is something that we considered seriously, and in fact, we initially dismissed the hyper-condensation of the SSB^f^-ssDNA fiber based on this interpretation; however, after careful examination of both the fluorescent intensity of the SSB^f^-ssDNA fibers during the salt jump transitions, and (even more convincingly) the force-spectroscopy with wild-type, unlabeled ssDNA, we came to the conclusion that protein dissociation was negligible or non-existent in all conditions when free SSB is present, except 750 mM NaOAc, and that dissociation could not fully account for our observations.

When natural ssDNA substrates (e.g., ssM13) are used, SSB does exhibit a larger apparent site size. In the example cited by the reviewers, at 300 mM NaCl, a site size of 77 nts is observed instead of 65 nts. There are two ways to interpret this:

1) that the actual, physical footprint of the protein is different for natural polynucleotides (i.e., that the molecules simply wrap differently, possibly with bulges), or

2) that the larger site size represented unoccupied ssDNA, thereby reflecting a fully wrapped, 65 nt binding mode, while 15% of the ssDNA is occluded owing to secondary structure formation. These need not be mutually exclusive, but the second interpretation is the one that is favored in the literature, and we agree with this interpretation of ensemble, equilibrium binding experiments.

As stated above, we initially shared the reviewers’ concerns that the greater degree of condensation observed in the TIRF experiments might be due to formation of secondary structure, and this was stated (both in the revised and original submission) as our reasoning for using force spectroscopy (magnetic tweezers) with wild-type, unlabeled SSB. This assay is extremely sensitive, and allowed us to measure the condensation of the SSB-ssDNA fiber (on a long, natural ssDNA substrate) as a function of both force and salt perturbation. When we compared force-extension curves for either ssDNA or SSB-coated ssDNA, we noted that a dramatic difference is apparent in the hysteresis of the curves. We interpret the absence of hysteresis in the presence of SSB as strong evidence that long-range DNA secondary structure is not capable of excluding SSB and contributing to condensation until the concentration of NaOAc was increased to 750 mM. This is readily apparent when we calculate the ΔΔE (energy) by subtracting the ssDNA-alone relaxation curves (which do not include the energetics of long-range secondary structures; see subsections “Force spectroscopy of single molecules of ssDNA and SSB-coated ssDNA reveals a nearly complete relief of hysteresis in SSB-ssDNA unfolding transitions” and “The SSB-ssDNA complex is a nearly isoenergetic landscape, relative to unstructured ssDNA, at physiological salt concentrations” for discussions and references) from the SSB-coated ssDNA curves at each salt concentration, where we see a nearly uniform binding energy, with the sole exception of the 750 mM NaOAc condition (new Figure 5—figure supplement 2). Therefore, at least in our magnetic tweezers experiment where some force is always being exerted on the molecule owing to the bead tethering (in contrast to the ensemble equilibrium binding described in the paragraph above), no measureable long-range secondary structure forms in the presence of SSB.

Based on the sum of our observations, i.e., the absence of noticeable intensity changes in the SSB-ssDNA fiber and the good agreement between our TIRF data and our magnetic tweezer experiments, we think that higher-ordered condensation of the SSB-ssDNA complex is the most correct explanation for our observations. The model that we propose, where SSB-mediates this condensation via SSB-SSB interactions or by engaging in multiple SSB-ssDNA interactions within single tetramers, rather than ssDNA hybridization, is largely warranted based on the complex properties of SSB, but it is a model nonetheless. We have expanded our Discussion to consider other possible explanations mentioned by the referees.

That said, in the condition where free SSB is omitted from our experiment, as in Figure 2 and Figure 2—figure supplement 1, we do observe protein dissociation during the time needed to measure condensation of the SSB-coated ssDNA fiber. We therefore conclude that the structural re-arrangements that we observe and the relatively constant amount of protein bound to the ssDNA, is normally driven or maintained through reversible mass action with free SSB.

*The major points are listed below*:

*1) The manuscript addresses the transitions in binding modes as a function of salt concentrations and the reviewers don't understand how the present data can be reconciled with published similar measurements albeit with different tools that the authors mention. In these previous studies such dramatic compaction was not reported. This was confusing and perhaps a discussion clarifying the anticipated reasons for the differences could be made*.

*5) As the authors mention, the extent of ssDNA compaction observed by both EM (Griffith) and AFM (Harmon et al.) studies is consistent with the extent of compaction expected from the SSB binding mode transitions. Compaction as measured by sedimentation studies (*[12]
*JBC 263, 4629) show that compaction tracks with occluded site size, when both measurements are performed with poly(dT). It seems that the additional ssDNA compaction suggested in the current study should be observable also by these techniques, but none has been reported. Why*?

Given the similarity of these comments, we address them together for brevity.

Neither the EM or AFM studies explored a significant range of salt concentrations, and in fact, were rather limited in the range of conditions tested. Perhaps more importantly, both methods require experimental conditions that physically “spread-out” DNA or protein-DNA complexes. Consequently, they are in fact quite perturbing of structures that are held together by weak interactions. Furthermore, if the spreading is not successful, then neither EM nor AFM studies will publish such structures because they appear as condensed, amorphous “blobs”. Hence, we offer these two reasons as possible explanations for why condensation was not reported earlier.

In the case of the EM papers, the experiments were designed to evaluate the morphology of the complexes as a function of SSB saturation and/or RecA filament-mediated displacement of SSB. The Hamon paper explored a rather limited range of salt concentrations, primarily focusing on developing a method to adsorb and spread SSB-ssDNA complexes onto mica for imaging, reporting contour lengths for only two conditions (20 mM TrisHCl (7.5), 20 mM NaCl and 50 uM spermidine or 20 mM TrisHCl (7.5), 300 mM NaCl, and 300 uM spermidine). Interestingly, they do show images of SSB-ssDNA complexes formed in 12 mM MgCl_2_ in their Figure 1, and these complexes form “highly condensed complexes that do not spread well*”*, consistent with our observations of strong, Mg(OAc)_2_-dependent compaction, but as a consequence of this highly condensed structure, they do not report measurements of these complexes. We also note that such condensed complexes can be easily dismissed by critics of AFM as being structures that aggregated on the ionic mica surface and perhaps “artifacts”. We did attempt to use AFM to image condensed complexes; however, we ran into two technical problems. We verified that conditions permitting condensation similarly resulted in poorly spread complexes that could not be used to measure contour lengths and resulted in poor imaging conditions owing to salt and aggregates on the mica. Removal of the salt requires rinsing the samples, which obviously perturbs the intramolecular condensation of the molecules, and we know from our TIRF experiments that condensation, decondensation and remodeling are rapid. Given these technical problems – as well as the complication of requiring spermidine for surface adsorption – we reasoned that the force spectroscopy experiments would be the most accurate method for measuring contour lengths at increasing salt concentrations using wild-type unlabeled SSB.

We cannot make a direct comparison to [12] because their sedimentation studies were done as a function of Mg^++^ concentration; in addition, their concentration of SSB was sub-saturating (either 101 or 87 nts per tetramer). However, upon scrutinizing the older literature, we discovered that in largely underappreciated work, [54] examined the sedimentation of SSB-coated ssDNA using native M13 ssDNA under conditions where the ssDNA was both under- and super-saturated. When excess SSB was present (as in our conditions), they observed a log-linear relationship between the sedimentation coefficient and salt concentration. We have extracted the values from their paper, and we have plotted those values against the length measurements from our single-molecule assays. We think that it is obvious that the log-linear trend is the same, indicating that our observations and method are in fact in good agreement with Schaper et al (Figure 6).

Author response image 1.**DOI:**
http://dx.doi.org/10.7554/eLife.08646.024

*2) Seeing is believing, but given the low spatial resolution of the methods certain structural assumptions need to be made with regard to fiber dimensions and as such compaction calculations are perhaps potentially in error? Is the model put forward that the compaction is due to long range interactions between different SSB tetramers the only one that can explain the data*?

We refer the reviewers to our general response above. However, “low spatial resolution” depends on the point of reference and the structural scale of the conclusion. The resolution in our TIRF experiment is ∼200-300 nm; however, the resolution in our magnetic tweezer experiments is on the order of a few nanometers (precise resolution changes with the amount of force applied).

Certainly several structural models could be advanced, including folding of the fiber into solenoid or fractal structures; however, our experiments are not informative as to what these structural features are. In the Discussion section we discuss that the molecular interactions could be explained by either tetramer-ssDNA interactions (for example, if tetramers do not strictly bind to ssDNA only contiguously, but rather as two or more discrete binding entities, similar to the tetramer bound to two molecules of 35-mer in the crystal structure), or through tetramer-tetramer interactions. Alternatively, long-range interactions could be explained by re-annealing of the ssDNA as secondary structure forms. Because the magnetic tweezers experiments allow us to directly measure secondary structure formation, which is apparent in the hysteresis when pulling vs relaxing curves are compared, we are able to directly eliminate this latter explanation for all of our conditions except when 750 mM NaOAc was used.

The model that we advance where fiber folding is mediated by either (presumably many and weak) SSB-ssDNA or SSB-SSB interactions is supported by the following:

A) Dimerization of SSB tetramers (i.e., formation of octamers) is a well-established phenomenon (14).

B) SSB diffuses on ssDNA rapidly and undergoes intersegmental transfer (Roy et al., Nature (2009), Zhou et al., Cell (2011) & Lee et al., JMB (2014)) across long distances.

C) SSB tetramers can directly transfer between ssDNA molecules without proceeding through a free-protein intermediate (Kozlov & Lohman, Biochemistry (2002, a&b); thus, this transient intermediate may be iteratively sustained along a long DNA molecule, and can contribute to the overall folding of the SSB-ssDNA fiber.

*3) The analogies to nucleosome condensation with salt changes is intriguing but the physiological relevance of this phenomenon has yet to be established for chromatin. While osmolarity changes in a prokaryote cell should be significant are they in the range used by Bell et al. for* E. Coli *in its natural environment? Perhaps some references here are appropriate.*

The intracellular salt concentrations of *E. coli* are very well documented using careful analytical methods by Tom Record’s lab. As stated by Richey et al. [in “Variability of the intracellular ionic environment of Escherichia coli. Differences between in vitro and in vivo effects of ion concentrations on protein-DNA interactions and gene expression.” J Biol Chem 262(15): 7157-7164, (1987)], “As the osmolality of the growth medium is varied from 0.1 to 1.1 osmolal, the total intracellular concentration of K^+^ increases linearly from 0.23 to 0.93 molal and the total intracellular concentration of glutamate increases linearly from 0.03 to 0.26 molal.” The biological significance of the condensation phenomenon we report is consequently physiologically relevant to changes in intracellular salt concentrations due to external changes in osmolality that are experienced by *E. coli*. This is further discussed in the subsection “The extent of intramolecular condensation of SSB-coated ssDNA exceeds expectations based on simple wrapping or binding-mode transitions” of the Results section, explaining our reasoning for performing experiments with potassium glutamate and magnesium ion. There is a long history on this important subject, which we cannot review in our manuscript; the interested reviewer is also referred to work by Wolf Epstein’s lab: e.g., [16]. “Cation Transport in Escherichia coli: V. Regulation of cation content.” J Gen Physiol 49(2): 221-234 and [17]. “Cation transport in *Escherichia coli*. VI. K exchange.” J Gen Physiol 49(3): 469-481.

We have significantly reduced our discussion of nucleosomes.

*4) It is well established that* E. coli *SSB can bind single stranded DNA in multiple binding modes that differ in the extent of wrapping of the ssDNA around the tetramer. In experiments with poly(dT), binding modes with occluded site sizes of ∼35, 40, 55 and 65 nucleotides per tetramer have been identified. The relative stability of these modes can be modulated in vitro by changing the salt concentration. As the salt concentration is increased, the mode of binding shifts from the 35 mode to the higher site size modes, resulting in progressive compaction of the ssDNA. These different binding modes also display very different cooperative behavior. The binding mode transitions have also been demonstrated by sedimentation, EM and AFM measurements. The previously reported occluded site size measurements as a function of salt type and concentration have generally been performed using poly(dT) in order to eliminate complications due to secondary structure (hairpins, etc.) that will form in natural ssDNA, such as M13 or lambda DNA that is used in the current study. In the original report of salt-induced SSB binding mode transitons (*[38]*) the apparent site sizes measured using poly(dT) were compared with measurements using ss M13 phage DNA. These showed that the apparent site size is severely overestimated when M13 ssDNA is used and that the discrepancy increases as the salt concentration increases (e.g., 65 nt for pol(dT) vs. 77 nt for M13 DNA at 0.3 M NaCl). This difference will be greater at higher salt concentrations. This difference is due to the fact that SSB, especially at high salt concentrations is unable to remove all of the secondary structure from natural ssDNA. This is partly because SSB does not bind with high cooperativity in the 55 or 65 modes. Therefore, the uncertainty in predicting what the expected compaction of ss lambda DNA based on site sizes measured on polydT makes the main conclusion difficult to assess. The authors need to address this before the manuscript can be published*.

*6) Bell et al. use minimum and maximum dimensions of SSB of 6 nm and 8 nm for the calculation of the “expected” extent of ssDNA compaction, but do not explain how they arrived at these estimates. Reviewers agree with the 8 nm estimate, which is from the tips of the L45 loops in the crystal structure. However, the other dimension from the crystal structure can be as small of 4.5 nm. When the calculation is done using 4.5 nm, the predicted length lower limit is lowered by 25%, which encompasses all of the data points except for the two at the highest salt concentrations. These two points are subject to the caveats mentioned elsewhere in the review*.

The SSB tetramer is roughly an ellipsoid of the dimensions 5 x 6 x 8 nm, with a surface groove (or ‘cinch’) around the waist that can be as small as 4 nm. After re-examining the SSB structure, we agree that we over-estimated the smallest possible axis. Nonetheless, as one reviewer pointed out in comment 2, these calculations make certain assumptions about the fiber conformation. Consequently, for reasons explained above, we have removed the figure, and instead refer to the relative length changes observed by AFM in Hamon et al.

*7) Bell et al. measured the SSB-induced ssDNA compaction using mixed sequence ssDNA (lambda), whereas the site size measurements are made using poly(dT). Hence, some of the “unexpected” additional compaction is certainly due to unmelted secondary structure in the lambda DNA. How much is not clear. The difference in site sizes measured using poly(dT) vs. natural ssDNA (mixed sequence) will be greater at the higher salt concentrations (> 0.3 M), consistent with the authors reporting the largest “additional unexpected” compaction at these higher salt concentrations*.

We think that this comment also relates to both of the major issues that were noted and addressed in our general response above. In addition, we now also provide intensity measurements for the molecules shown in Figure 1 (see Figure 1—figure supplement 1) and also provide the intensity measurements for all molecules summarized in Figure 3, plotted as a scatter plot as a function of salt concentration in Figure 3—figure supplement 3.

*8) Potential SSB dissociation. The authors initially state that the amount of SSB bound to the ssDNA during the collapse transition is “essentially constant”, attributing the loss of SSB fluorescence to photobleaching. Yet they also show that at the very high salt concentrations (> 750 mM), some of the SSB does dissociate, although this is not considered to be a major factor. The question is how much dissociates and is it reasonable to dismiss this? SSB does not bind as tightly to natural ssDNA as it does to poly(dT), hence this is likely not inconsequential. Please explain*.

It is not our intention to dismiss the possibility of SSBf dissociation, especially in the experiment presented in Figure 2, where free SSB is not available. In fact, we think it is quite clear that the fluorescent SSB does dissociate under those conditions; however, all other experiments were performed under conditions where free SSB was available to rebind before, during, and after length transitions that coincide with salt and/or force perturbations. We did explicitly state that there would be dissociation at 750 mM, but we now explicitly excluded this condition from our discussions and we note that all of our conclusions remain valid even if this concentration is excluded. We have further analyzed the images in Figure 2 to quantify the dissociation in the absence of free protein.

*9) Could the constant fluorescence intensity observed upon shortening of the molecule (*Figure 1*) be the result of altering the environment of the fluorophore? As indicated, this result is surprising given that increasing the salt should theoretically decrease the number of available binding sites on the ssDNA*.

As we indicate in the text, the single-molecule experiments were performed with SSB labeled with AF488, while the equilibrium titrations were performed with SSB labeled with 5,6-carboxyflourescein. This is because AF488-labeled SSB is brighter and does not photobleach as quickly as fluorescein-labeled SSB, and is less sensitive to environmental changes induced by binding. Therefore, AF488-labeled SSB is most well suited for single-molecule measurements, and fluorescein-labeled SSB is more suitable for traditional, ensemble measurements. In the absence of salt, we have observed a 2.6-fold increase in fluorescence upon polydT binding to SSB-AF488, but only a 2.2-fold fluorescence enhancement in the presence of salt (200 mM NaOAc and 5 mM Mg(OAc)_2_). Therefore, the AF488-labeled ssDNA is actually slightly quenched (∼15%) in its fluorescence rather than enhanced upon the transition to higher salt conditions. For a ‘salt jump’ from 100 to 250 mM (as shown in Figure 1), the expected site size change for M13 ssDNA should be, per [38], 62 nt at 100 mM to 77 nt at 300 mM, corresponding to a ∼24% change to slight higher salt concentration, and yet, we observe a 40% change in length, with no change in total fluorescence, but rather a concentration of the fluorescence as the fiber condenses. Also, please see our response to the following question.

*Is constant fluorescence observed when no free SSB is included in the flow (*Figure 2*)*?

No, the fluorescence decreases. We have included a new Supplement to Figure 2 (Figure 2—figure supplement 1), where we have analyzed and quantified the intensity of the molecules shown in Figure 2. This figure shows that 20% of the SSB dissociates in the 0 to 100 mM NaOAc transition, and 60% of the protein dissociates in the 0 to 400 mM NaOAc transition. When the salt concentration decreases back to 0 mM, we do not observe any additional intensity change, indicating that the single-molecule measurement with SSB-AF488 is insensitive to the molecular environmental changes coinciding with condensation and de-condensation.

*Additionally, we are missing the connection for how this result led to the authors asking whether condensation is reversible (subsection “Single molecules of SSB-coated ssDNA reversibly condense in response to increasing salt concentration”. “We therefore asked”) – clarification on the reasoning as well as a brief mention of possible explanations would be beneficial in this section*.

We have added clarification to the subsection “Single molecules of SSB-coated ssDNA reversibly condense in response to increasing salt concentration” of the revised manuscript.

*10) Is the salt-induced condensation reversible with free protein and not just in the absence of it (*Figure 2*)*?

Yes, this is most obvious in the force-extension traces, but it is also true of the experiments performed in Figure 1 and throughout the rest of the manuscript.

*11) Clarify how the values plotted in*
Figure 3
*were obtained – was a specific nt/SSB ratio used or were titrations done as in 3A for all of the [NaOAc] in 3B*?

The fold-increase in Figure 3 is from the amplitude from the titrations performed in Figure 3; however, a larger number of titrations were performed than are represented in 3A in order to prevent 3A from being overcrowded. In other words, the fold-increases in Figure 3 were determined from a full stoichiometric titration where each titration was completely and fully saturated.

We also performed a “salt back--titration”, where complexes were formed in the absence of salt, and salt was increased in order to determine the concentration at which SSB^f^ fully dissociated from ssDNA. This data is now included as a Figure 3—figure supplement 1.

*12) In the subsection “Force spectroscopy of single molecules of ssDNA and SSB-coated ssDNA reveals a nearly complete relief of hysteresis in SSB-ssDNA unfolding transitions”, second paragraph, it is mentioned that the modest hysteresis at high [NaOAc] is consistent with the interpretation that the decrease in fluorescence enhancement with SSBf at high [NaOAc] is due to partial dissociation. This is the first time that this interpretation and the fact that fluorescence enhancement decreases at high [NaOAc] is mentioned – a brief discussion should be provided earlier within the Results section referring to*
Figure 3.

We had already mentioned this in the subsection “The extent of intramolecular condensation of SSB-coated ssDNA exceeds expectations based on simple wrapping or binding-mode transitions”, third paragraph. There, we articulated our caveat that the greater than expected condensation at high [NaOAc] could be due to dissociation of SSB^f^ and formation of secondary structure. Nonetheless, we have tried to more accurately articulate this possibility throughout the manuscript.

*13) The figure description for*
Figure 5
*is difficult to follow. In the figure it appears as though purple and gray represent a single area (separate from light purple), however the description seems to be written as if they are all separate*.

We have updated the legend to the “purple and gray striped area” instead of the “gray area” and clarified the description.

*14) The discussion of nucleosomes and chromatin seems unnecessary - the statement “our observation that SSB...during DNA metabolism” (Discussion, third paragraph) would be sufficient and can be added on to the end of the preceding paragraph*.

We have taken the reviewers’ recommendations and we have shortened and edited this discussion accordingly.

*15) The suggestion in the Discussion that SSB-interacting partners might modulate the condensation state of the SSB-ssDNA fiber by altering the binding mode is supported by work on the interaction between SSB and both PriA and PriC (*[6]*,*
[63]*). Both proteins have been shown to induce the 35nt binding mode of SSB in a manner dependent upon complex formation*.

We have added these references, and have included this possibility to the Discussion.